# General method for carbon–heteroatom cross-coupling reactions via semiheterogeneous red-light metallaphotocatalysis

Geyang Song[1,2], Wei Zhang[1,2], Jiameng Song[1], Qi Li[1], Yuyu Feng[1], Hongyu Liang[1], Tengfei Kang[1], Jianyang Dong[1], Gang Li[1], Juan Fan[1], Xue-Peng Zhang [1], Quan Gu [1]✉, Chao Wang [1] & Dong Xue [1]✉

Combining transition-metal catalysis with photocatalysis has emerged as a valuable, complementary approach for achieving carbon–heteroatom cross-coupling reactions. However, the need to use blue or high-energy near-UV light leads to problems with scalability, chemoselectivity, and catalyst deactivation, which have limited the synthetic applications of this combination. Herein, we report a method for red-light-driven nickel-catalyzed cross-coupling reactions of aryl halides with 11 different types of nucleophiles using a polymeric carbon nitride (CN-OA-m) as a photocatalyst. This semiheterogeneous catalyst system enabled the formation of four different types of carbon–heteroatom bonds (C–N, C–O, C–S, and C–Se) with a wide range of substrates (more than 200 examples) with yields up to 94%. Moreover, the photocatalyst is easily recovered and recycled, which makes it a promising new tool for the development of other reactions involving red-light metallaphotoredox catalysis.

Carbon–heteroatom cross-coupling reactions catalyzed by Pd[1,2], Cu[3], Ni[4], and other metals[5–10] offer an efficient method for the synthesis of structurally complex pharmaceuticals, pesticides, functional materials, and fine chemicals[11–15]. However, the success of these reactions heavily depends on the selection of appropriate parameters, including the ligand (e.g., phosphines, carbenes, or oxamides), precatalyst, base, additive, solvent, and reaction temperature, as well as the electronic and structural properties of the substrates[16–24]. Thus, case-by-case optimization of the reaction conditions is often required. Combining transition-metal catalysts with well-established photocatalysts such as metal complexes, organic dyes, and organic semiconductors has emerged as a valuable approach to overcoming some of the above-mentioned challenges (Fig. 1A)[25–33]. However, the light-promoted reactions reported to date require blue or high-energy near-UV light,

which lead to problems with scalability, chemoselectivity, and catalyst deactivation[33] that result from competitive absorption of light by the substrates and intermediates. Therefore, the synthetic utility of the combination of transition-metal catalysis and photocatalysis is currently limited[34–38]. Recently, the Rovis group pioneered red-light-driven C–N cross-coupling reactions mediated by the combination of a metal catalyst and an osmium photocatalyst (Fig. 1B)[39–41]. These investigators found that a dual Ni/Os catalyst system has good substrate compatibility and is easy to scale up, showcasing the potential utility of red-light-driven coupling reactions[42]. This study shows that red light can enhance the desired reactivity by inhibiting unwanted photodegradation pathways (which are usually caused by the use of blue or high-energy near-UV light), which may provide new ideas for breaking through bottlenecks in the development of cross-coupling

[1]Key Laboratory of Applied Surface and Colloid Chemistry of Ministry of Education, School of Chemistry and Chemical Engineering, Shaanxi Normal University, Xi'an 710119, China. [2]These authors contributed equally: Geyang Song, Wei Zhang. ✉e-mail: guquan@snnu.edu.cn; xuedong_welcome@snnu.edu.cn

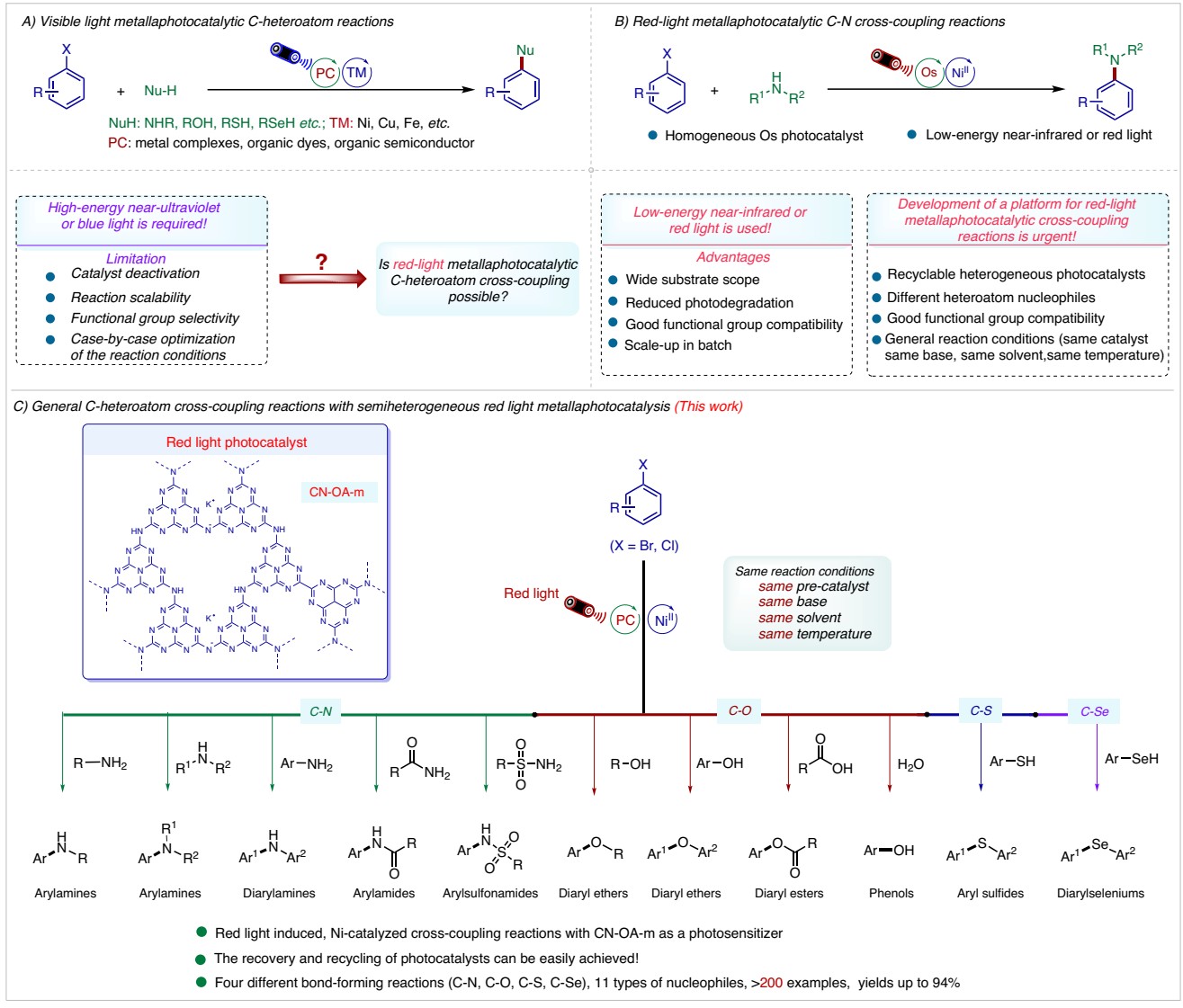

**Fig. 1 | Development of photocatalysis for C(sp²)–heteroatom bonds cross-couplings. A** The homogeneous photocatalysts for C(sp²)–heteroatom cross-couplings formation between aryl halides with different nucleophiles. **B** The Red-light metallaphotocatalytic C-N cross-coupling reactions. **C** This work. General C-heteroatom cross-coupling reactions with semiheterogeneous red light metallaphotocatalysis.

reactions. Despite some successes in the field[42–48], the lack of inexpensive, readily available, versatile, recyclable red-light catalyst systems has hindered the development of red-light metallaphotoredox catalysis.

Heterogeneous semiconductors such as carbon nitride ($C_3N_4$) are promising alternatives to other photocatalysts because the former are easy to prepare and can be recovered by filtration or centrifugation[49–54]. Although these semiconductors have been applied to transition-metal-catalyzed coupling reactions[29–31,33,55–74], they have rarely been used for red-light-catalyzed cross-coupling reactions. In this context, we report a method for red-light-driven Ni-catalyzed cross-coupling of aryl halides with 11 different nucleophiles using the polymeric carbon nitride CN-OA-m as a photocatalyst under the same reaction conditions (same pre-catalyst, temperature, solvent, and base, Fig. 1C). This versatile, semiheterogeneous catalyst system enables the formation of four different types of carbon–heteroatom bonds (C–N, C–O, C–S, C–Se) and exhibits a broad substrate scope (more than 200 examples, and with yields up to 94%).

## Results and discussion

### Investigation of the reaction conditions

In 2017, Zhang et al. reported the synthesis of CN-OA-m from urea and oxamide in a molten salt[75]. The conduction band potential of this polymer is at −1.65 V vs Ag/AgCl, and the valence band potential is at 0.88 V vs Ag/AgCl[33], and an obvious absorption band in the 460–700 nm region[75,76]. These investigators also demonstrated that CN-OA-m can absorb visible light at wavelengths of >500 nm and exhibits high hydrogen-production activity. For our initial experiments, we selected 3,5-dimethylbromobenzene (**1**) and pyrrolidine (**2**) as model substrates to optimize the conditions for the desired coupling reaction with CN-OA-m as a photocatalyst (Table 1)[77]. Target amination product **3** was obtained in 91% isolated yield upon red-light irradiation (660–670 nm) of the substrates under argon at 85 °C for 24 h in dimethylacetamide (DMAc) containing NiBr₂·glyme and 1,4,5,6-tetrahydro-1,2-dimethylpyrimidine (*m*DBU) (entry 1). Other photocatalyst–carbon nitride ($C_3N_4$, entry 2)[78], mesoporous graphitic carbon nitride (mpg-$C_3N_4$, entry 3)[79], P-doped carbon nitride prepared from diammonium hydrogen phosphate and urea (p-$C_3N_4$,

## Table 1 | Optimization of the reaction conditions

| Entry | Variation from standard conditions | Yield (%)[a] |
|---|---|---|
| 1 | None | 94 (91[b]) |
| 2 | $C_3N_4$ instead of CN-OA-m | 71 |
| 3 | mpg-$C_3N_4$ instead of CN-OA-m | 51 |
| 4 | p-$C_3N_4$ instead of CN-OA-m | 43 |
| 5 | g-$C_3N_4$ instead of CN-OA-m | 46 |
| 6 | RP-$C_3N_4$ instead of CN-OA-m | 76 |
| 7 | MC-$C_3N_4$ instead of CN-OA-m | 83 |
| 8 | blue LEDs (460–465 nm) | 31 |
| 9 | green LEDs (525–530 nm) | 56 |
| 10 | white LEDs (6500 K) | 62 |
| 11 | light (660-670 nm), rt. | NR |
| 12 | no $m$DBU | 13 |
| 13 | no light | NR |
| 14 | no light, 120 °C | NR |
| 15 | no Ni catalyst | NR |

[a]Standard conditions: 3,5-dimethylbromobenzene (**1**, 0.2 mmol), pyrrolidine (**2**, 2.0 equiv., 0.4 mmol), NiBr$_2$·glyme (10.0 mol%), CN-OA-m (10 mg mL$^{-1}$), $m$DBU (0.3 mmol), DMAc (1 mL), Ar. Yields were determined by $^1$H NMR spectroscopy with 1,3-benzodioxole as an internal standard. NR = no reaction; rt = room temperature.
[b]Isolated yield. For details, see Supplementary Information.

entry 4)[80], graphitic carbon nitride (g-$C_3N_4$, entry 5)[81], RP-$C_3N_4$ (a red polymeric carbon nitride containing Na and K atoms; entry 6)[82], and MC-$C_3N_4$ (a reddish-brown polymeric carbon nitride containing Na, K, and Li atoms; entry 7)[83]—gave lower yields (43–83%) than CN-OA-m (see also Table S1). Our findings revealed that these organic semiconductors are efficient red-light photocatalysts. The solvent also had a significant impact on the reaction yield, with DMAc being the most effective (Table S3). Of the various tested Ni catalysts, NiBr$_2$·glyme exhibited the highest activity (Table S4). The wavelength of the light also played an important role, with 660–670 nm red light exhibiting the highest yield (compare entry 1 with entries 8–10, and see Table S6). The reaction was also sensitive to temperature; no C–N coupling product was obtained below 45 °C (entry 11, See SI, section 6). As the temperature increases, the yield increases significantly; however, once the threshold of 90 °C is surpassed, further increasing the temperature leads to negligible changes in yield (See SI section 6). These findings further underscore the importance of temperature control in this transformation. The choice of base also played a crucial role (entry 12); only organic bases were effective, with the mild, soluble base $m$DBU giving the best results (Table S7). Organic bases may serve two functions: one is to act as a base for deprotonation, and the other may involve serving as an electron donor to complete the photocatalytic cycle. By testing the oxidation potential of organic bases, it was revealed that the oxidation potential of $m$DBU is E$_{p/2}$ = + 1.39 V (Ag/AgCl in MeCN), which has a better matching oxidation potential (See SI section 13). Moreover, the reaction did not occur in the absence of light (entry 13), even at high temperatures (entry 14). Control experiments revealed that the reaction failed to proceed in the absence of a Ni catalyst (entry 15)[77].

### Substrate Scope of C–Heteroatom Couplings

With the optimized conditions in hand, we next investigated the scope with respect to the nucleophiles and aryl halides (Figs. 2, 3, 4). We found that 11 different types of nucleophiles efficiently coupled with 3,5-dimethylbromobenzene (**1**) or 4-bromobenzonitrile to afford the corresponding products in high yields. The reaction conditions were compatible with primary amines bearing a wide range of functional groups, including straight chains (**4** and **5**), a ketal (**6** and **8**), hydroxyl groups (**7** and **12**), a vinyl group (**9**), and a carbamate ester (**10**). Notably, no C–O coupling was observed for the amino alcohol substrates under the standard conditions (**7** and **12**). Cyclic primary amines also coupled efficiently (**10–14**), as did secondary amines (**16–19**). Notably, reactions of secondary aliphatic amines did not require an external organic base, perhaps because they can act as both a substrate and a base. Primary aliphatic (**20–26**), aryl (**27–29**), heteroaryl (**30** and **31**), and secondary aliphatic (**32–34**) amides also coupled successfully, providing N-aryl amides in 53–78% yields. In addition, sulfonamides with various substituents delivered the desired products (**35–45**) in 43–69% yields. Pyrazole also gave corresponding C–N coupling product **46** in 63% yield. Aniline, as an important substrate, undergoes a C–N cross-coupling reaction with 4,4-dimethyl-2,2'-bipyridine (d-Mebpy) as the ligand to produce the desired diarylamine product (**47–55**); however, without the ligand, the reaction hardly occurs.

Next, we also explored the reaction scope with respect to the aryl halides (Figs. 2, 3). Aryl halides containing various functional groups efficiently underwent C–N coupling with n-butylamine, delivering the desired aryl C–N coupling products in generally high yields. Electron-rich aryl halides with para substituents such as Me (**56**), Bpin (**57**), SMe (**58**), OMe (**59**), and OCHF$_2$ (**60**) exhibited high reactivity. Less reactive ortho-substituted aryl bromides also afforded the target products (**74** and **75**). Electron-deficient aryl halides were also suitable substrates for this red-light-driven C–N coupling reaction: products bearing a sulfoxide group (**63**), an unprotected sulfonamide (**64**), a ketone (**65** and **66**), a cyano group (**67**), a trifluoromethyl group (**68**), a halogen atom (**69**) could be obtained.

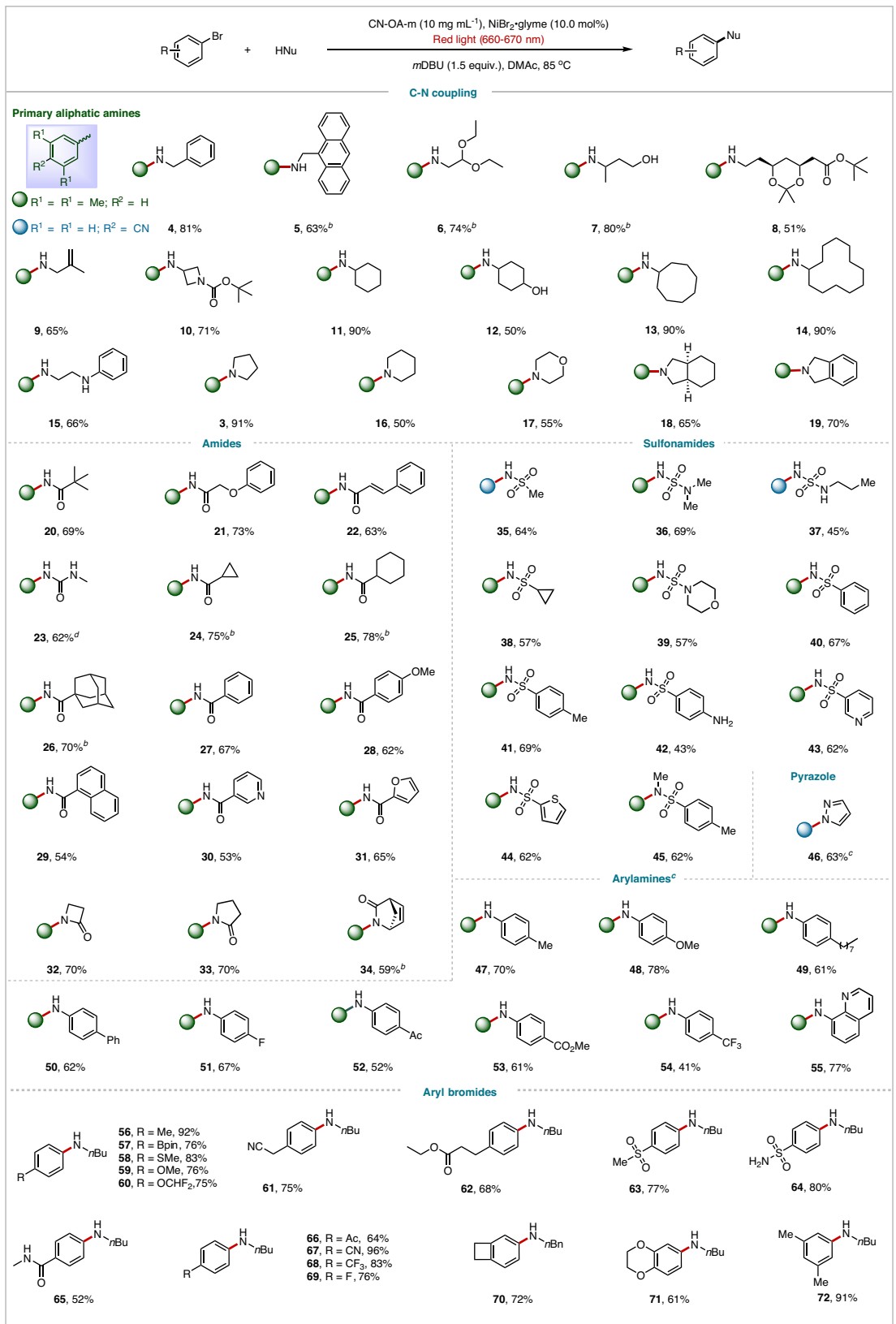

**Fig. 2 | C-N cross-coupling reactions of substrate scope.** [a] C-N cross-coupling reaction conditions, unless otherwise noted: aryl bromide (0.2 mmol), amine (0.4 mmol), CN-OA-m (10 mg mL⁻¹), NiBr₂·glyme (10.0 mol%), DMAc (1 mL), *m*DBU (1.5 equiv.), red light, 85 °C, Ar, 24 h. [b] 48 h. [c] with *d*-Mebpy as a ligand. Isolated yields are provided. For details, see Supplementary Information.

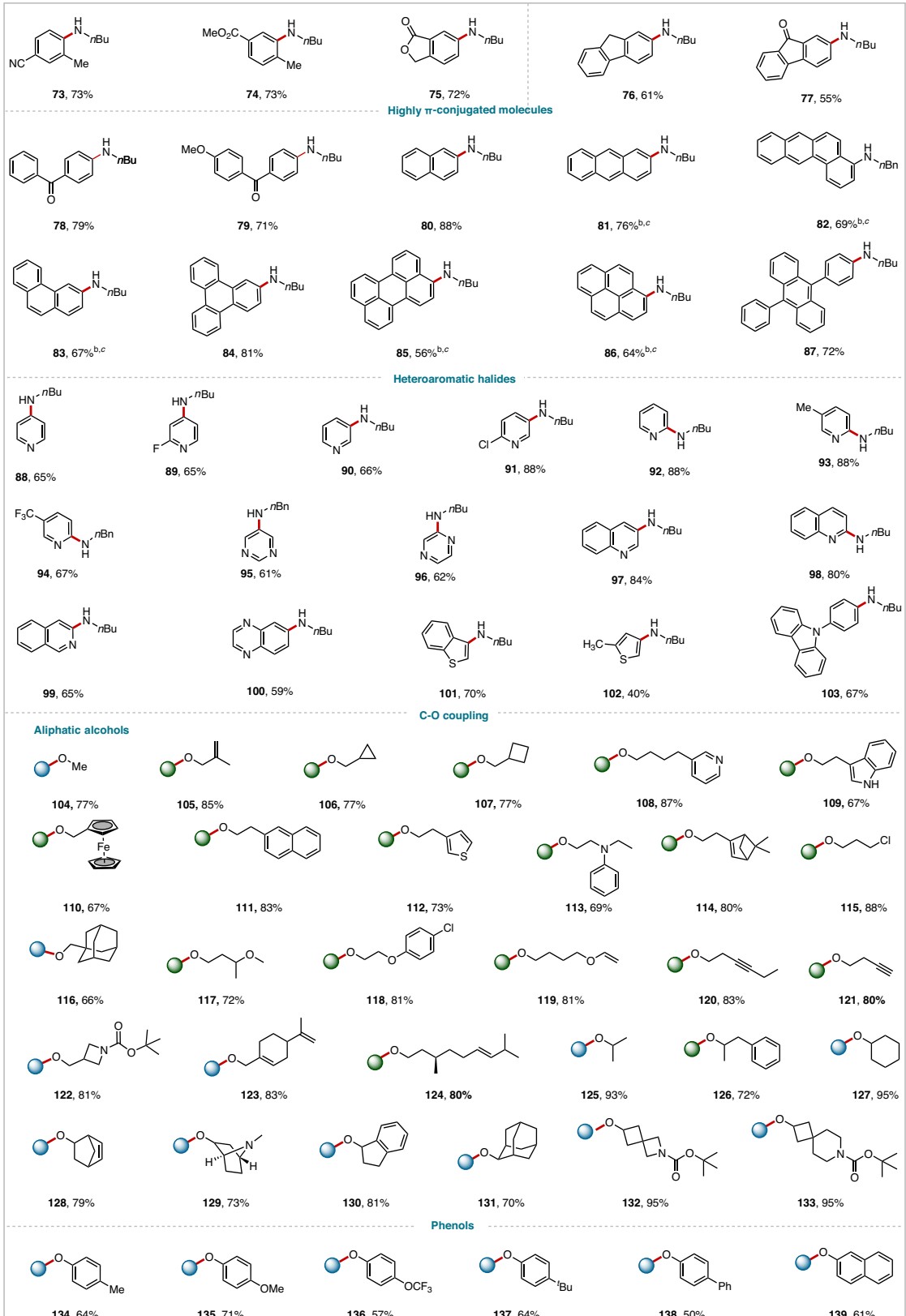

**Fig. 3 | C-N/O cross-coupling reactions of substrate scope.** [a] C-N/O cross-coupling reaction conditions, unless otherwise noted: aryl bromide (0.2 mmol), amine (0.4 mmol), CN-OA-m (10 mg mL⁻¹), NiBr₂·glyme (10.0 mol%), DMAc (1 mL), mDBU (1.5 equiv.), red light, 85 °C, Ar, 24 h. [b] 48 h. [c] with d-Mebpy as a ligand. [e]C-O/

S/Se cross-coupling reaction conditions, aryl bromide (0.2 mmol), alcohol (0.6 mmol), CN-OA-m (10 mg mL⁻¹), d-tbbpy (10.0 mol%), NiBr₂·glyme (10.0 mol %), DMAc (1 mL), mDBU (1.5 equiv.), red light (620-630 nm), 85 °C, Ar, 24 h. Isolated yields are provided. For details, see Supplementary Information.

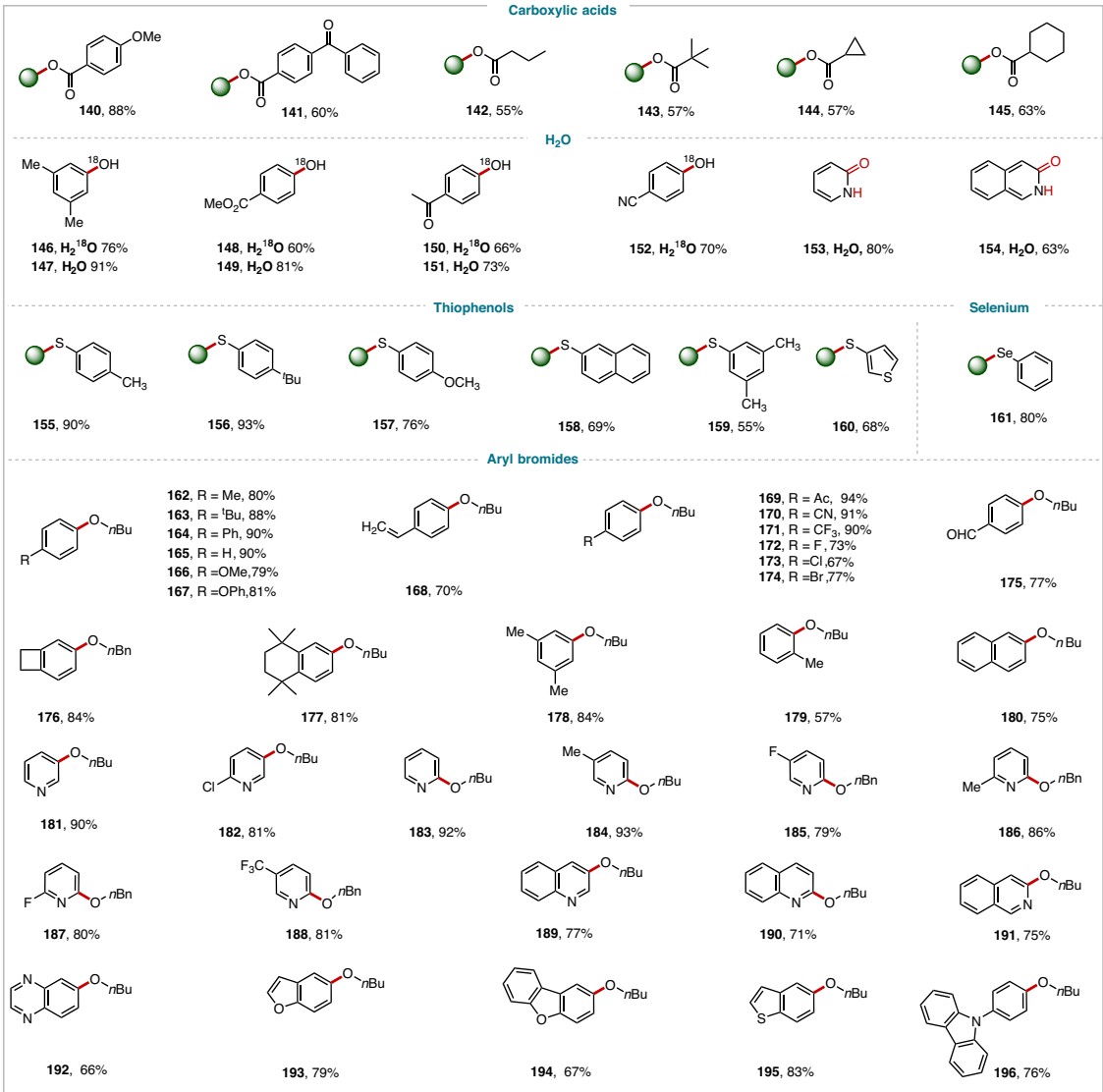

**Fig. 4 | C-O/S/Se cross-coupling reactions of substrate scope.** [a] unless otherwise noted: aryl bromide (0.2 mmol), alcohol (0.6 mmol), CN-OA-m (10 mg mL$^{-1}$), *d*-tbbpy (10.0 mol%), NiBr$_2$·glyme (10.0 mol %), DMAc (1 mL), *m*DBU (1.5 equiv.), red light (620-630 nm), 85 °C, Ar, 24 h. Isolated yields are provided. For details, see Supplementary Information.

Satisfactory yields were also obtained with multisubstituted aryl halides (**72–75**). Aryl halides with a series of highly π-conjugated and photosensitive substrates such as fluorene (**76**), 9-fluorenone (**77**), benzophenone (**78–79**), naphthalene (**80**), anthracene (**81**), phenanthrene (**82–83**), triphenylene (**84**), perylene (**85**) or pyrene (**86**) also afforded the desired products in satisfactory yields. Aryl halides containing a heterocycle—such as pyridine (**88–94**), pyrimidine (**95**), pyrazine (**96**), quinoline (**97–98**), isoquinoline (**99**), quinoxaline (**100**), benzothiophene (**101**), thiophene (**102**), or carbazole (**103**)—were transformed into the corresponding heteroarylamines. Such structures are important in pharmaceutical chemistry.

Further, we explored C-O/S/Se coupling reactions with nucleophiles other than nitrogen (Figs. 3 and 4). It should be emphasized that the coupling reactions of oxygen-containing, sulfur-containing, and selenium-containing substrates as nucleophiles are not observable under standard conditions without a ligand. However, by introducing bipyridine ligands, and adjusting the wavelength of red light to 620–630 nm (Table S13), the efficiency of the reaction is significantly improved, facilitating the smooth progression of the coupling process. Aliphatic alcohols underwent C–O coupling with 4,4-di-tert-butylbipyridine (*d*-tbbpy) as a ligand to afford the corresponding ether products (**104–133**), showcasing the practicality of this method. Other oxygen-containing nucleophiles—phenols (**134–139**), carboxylic acids (**140–145**), H$_2$O and H$_2^{18}$O (**146–154**)—were also good coupling partners, yielding the desired C−O coupling products in 50–91% yields. Notably, 2-bromopyridine and 3-bromoisoquinoline afforded the lactam products (**153–154**). Thiol coupling products **155–160** and selenium coupling product **161** were also obtained in satisfactory yields. Simultaneously, we explored the C-O coupling reaction of aryl halides with *n*-butyl alcohol (**162–196**), delivering the desired aryl C−O coupling ether products in generally high yields.

To explore further applications of our methodology, we attempted to expand the reaction scope to more affordable yet more challenging aryl chlorides. Given the stronger C-Cl bond, it was not unexpected that reactions yielded products in low yields under the conditions used for aryl bromides. However, after adding TBAI as an additive and extending the reaction time, satisfactory yields could be attained (Fig. 5). The reaction provided the cross-coupling products in good yields with aryl chlorides (**197–228**), containing electron-rich (Me, OMe) and electron-deficient (ester, ketone, nitrile) groups, as well as disubstituted aryl

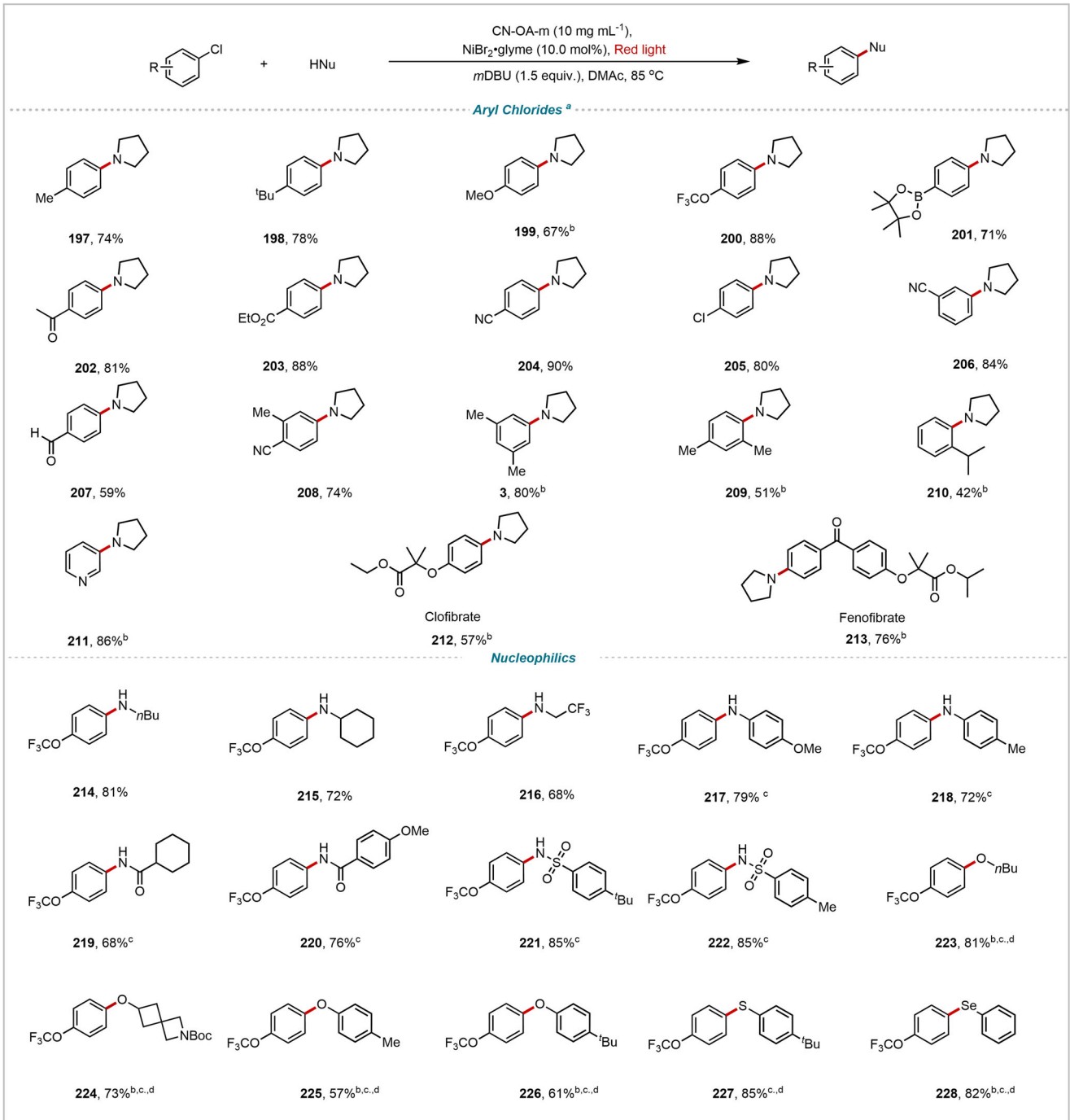

**Fig. 5 | Cross-coupling reactions of aryl chlorides.** [a] Reaction conditions, unless otherwise noted: aryl chloride (0.2 mmol), NuH (0.4 mmol), CN-OA-m (10 mg mL⁻¹), NiBr₂·glyme (10.0 mol %), DMAc (1 mL), *m*DBU (1.5 equiv.), red light (660–670 nm), 85 °C, Ar, 36 h. [b] 48 h. [c] with *d*-tbbpy (10.0 mol%). [d] 620–630 nm, NuH (0.6 mmol). Isolated yields are provided. For details, see Supplementary Information.

chlorides (**3**, **208**–**209**). Notably, it also showed excellent compatibility with the aldehyde group (**207**). Notably, more complex, biologically relevant substrates were also tolerated. For example, fenofibrate (**212**) and clofibrate (**213**) gave excellent yields. Meanwhile, a series of nucleophilic groups—including aliphatic amines (**214**–**216**), aromatic amines (**217**–**218**), amides (**219**–**220**), sulfonamides (**221**–**222**), alcohols (**223**–**224**), phenols (**225**–**226**), thiols (**227**), and selenol (**228**)—all provided the corresponding coupling products in excellent yields.

Then, we also investigated the C–heteroatom bond coupling of various bioactive molecules (Fig. 6). Specifically, the following bioactive molecules containing C(sp²)–Br bonds were efficiently

converted to the desired C–X coupling products: gemfibrozil (**229**), diacetonefructose (**230** and **237**), epiandrosterone (**231**), diacetone-d-glucose (**232**), tocopherol (**233** and **234**), celecoxib (**235**), l-menthol (**236**), stigmasterol (**238**) and epicholesterol (**239**). Notably, celecoxib bromide was compatible with the reaction conditions and could be used to obtain excellent yields of coupling products **240**–**262**, making our method highly attractive for the modification of bioactive molecules. These examples further demonstrate the broad applicability of this red-light-driven Ni-catalyzed cross-coupling reaction, highlighting its significant potential utility for the synthesis of drug-like molecules.

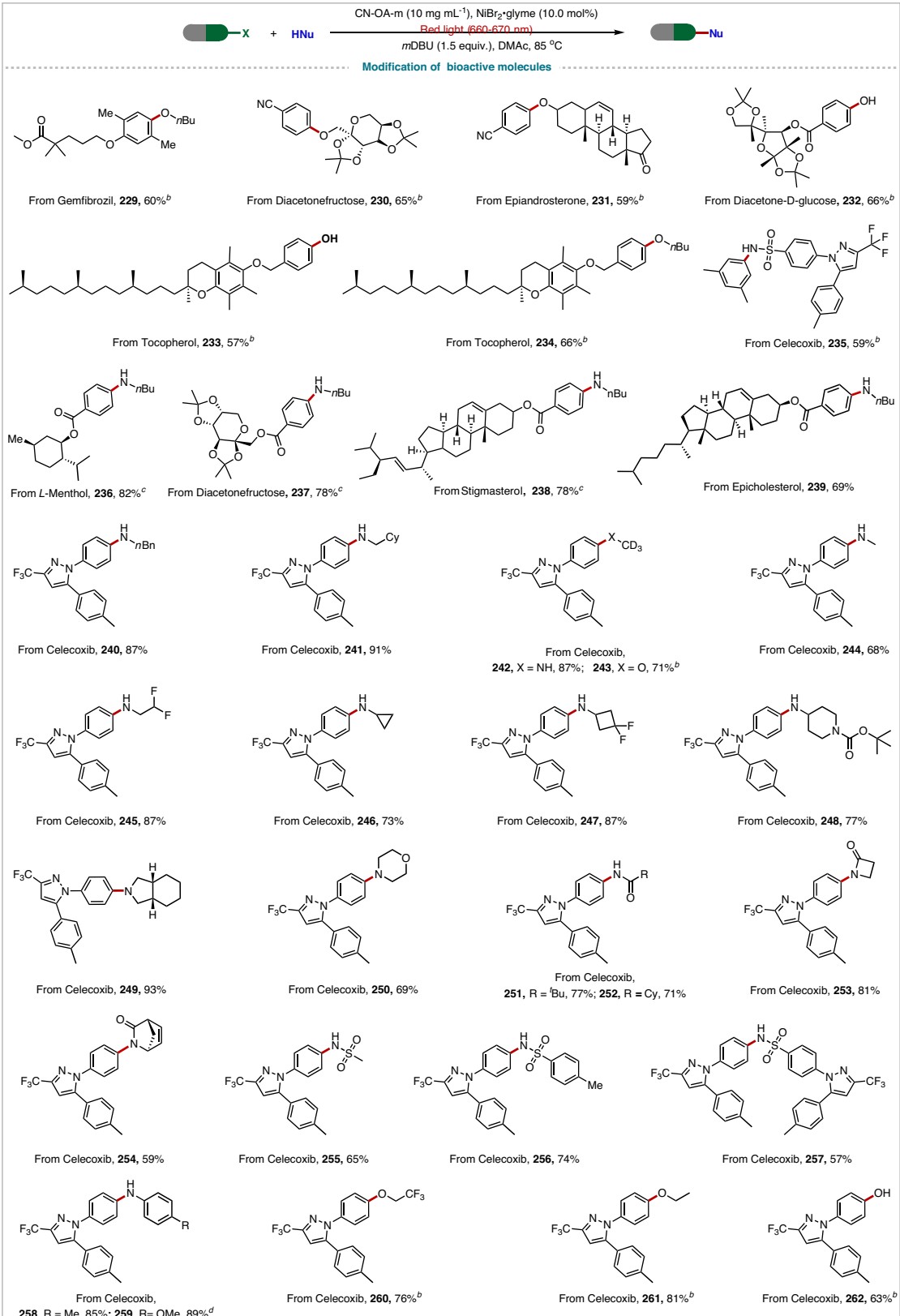

**Fig. 6 | Cross-coupling reactions of bioactive molecules.** [a] Brominated-bioactive molecule (0.2 mmol), amine (0.4 mmol), CN-OA-m (10 mg mL$^{-1}$), NiBr$_2$·glyme (10.0 mol%), DMAc (1 mL), $m$DBU (1.5 equiv.), red light (660-670 nm), 85 °C, Ar, 24 h. [b] CN-OA-m (10 mg mL$^{-1}$), NiBr$_2$·glyme (10.0 mol%), alcohol (0.6 mmol), $d$-tbbpy (10.0 mol%), red light (660-670 nm). [c] 48 h. [d] with $d$-Mebpy as a ligand. Isolated yields are provided. For details, see Supporting Information.

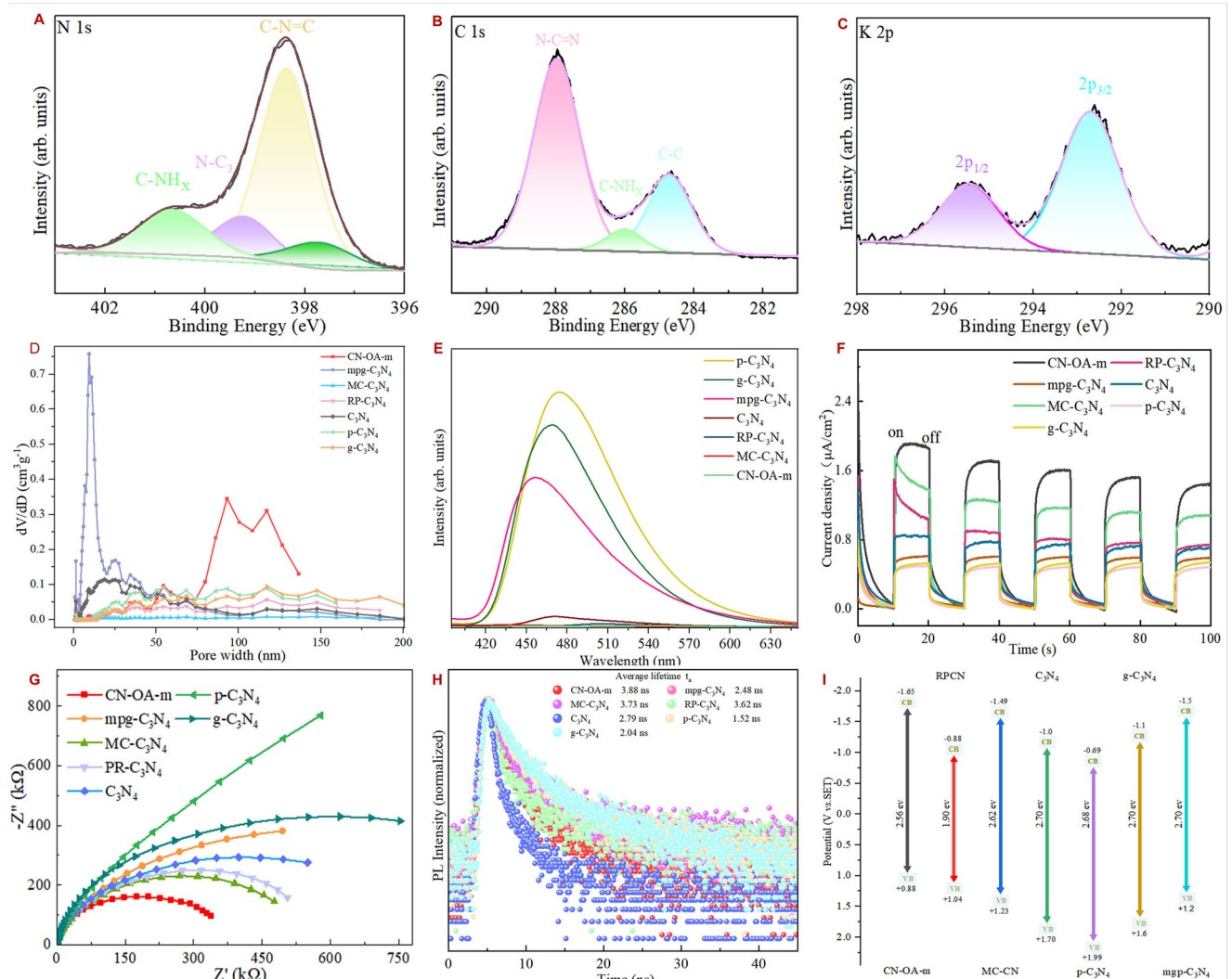

**Fig. 7 | Structure characterization of different C3N4. A–C** XPS spectra of CN-OA-m of C 1 s, N 1 s, K 2p; (**D**) The pore size distribution; (**E**) The photoluminescence (PL) spectra; (**F**) Photocurrent responses; (**G**) The electrochemical impedance spectra (EIS); (**H**) Time-resolved photoluminescence (TRPL) spectra; (**I**) Electrochemical potential diagrams.

## Structural characterization

In order to clarify the relationship between the structures of those catalysts and their activity in the reactions, a series of catalysts (C3N4, p-C3N4, g-C3N4, mpg-C3N4, RP-C3N4, MC-C3N4, CN-OA-m) were systematically investigated in terms of their physicochemical properties and structural features, including X-ray photoelectron spectroscopy (XPS), specific surface area, photoluminescence (PL), photocurrent responses, electrochemical impedance spectroscopy (EIS), time-resolved photoluminescence (TRPL) spectra, and band structures analyses. XPS analysis of CN-OA-m confirms the presence of C, N, and K elements on the surface (Figs. 7A–C, S18). The analysis of the N 1 s spectrum for CN-OA-m reveals three peaks at 399.2, 398.4, and 397.7 eV, which can be attributed to C–NH$_x$, N–C$_3$, and C–N = C, respectively. Additionally, a weak peak at 396.7 eV is observed, which is attributed to a negatively charged C = N–C moiety, potentially balancing the positive charge of K$^+$ ions. Furthermore, the C 1 s spectrum of CN-OA-m shows peaks at 288.0, 286.0, and 284.7 eV, which are attributed to N–C = N, C–NH$_x$, C–C bonds, respectively (Fig. 7A–C). These findings further suggest that defect energy levels can be generated between the conduction band and the valence band. Such defect levels can effectively trap photogenerated electrons, thereby suppressing the recombination of photogenerated electron–hole pairs

in CN-OA-m, which in turn promotes enhanced photoredox activity. The N$_2$ adsorption isotherm of all semiheterogeneous photocatalysts reveals that CN-OA-m signifying a highly developed porous network (Fig. S20). The Brunauer–Emmett–Teller (BET) surface area of CN-OA-m are 43.8 cm$^2$g$^{-1}$, with an average pore diameter of 13.80 nm. The BET pore size distribution of CN-OA-m exhibits the micro-meso-macropores characteristics (Figs. 7D, S21), with particularly significant proportion of macropores compared to other semiheterogeneous photocatalysts. The plentiful pores in CN-OA-m generate a high specific surface area, fully exposing catalytic sites while shortening diffusion paths for photo-induced electron–hole pairs, which collectively boosts its photocatalytic efficiency. By comparison, p-C3N4, g-C3N4, and mpg-C3N4 display much stronger photoluminescence signals, pointing to pronounced recombination of the photogenerated charge carriers. In contrast, the weakest photoluminescence (PL) intensity observed in CN-OA-m indicates suppressed carrier recombination (Figs. 7E, S22). This trend facilitates efficient electron transfer from the emissive state to Ni(II), potentially leading to the formation of Ni(I) species. The transient photocurrent response of different C3N4 electrodes were measured to gain in-depth insight into charge separation efficiency (Figs. 7F, S23). In contrast to the other C3N4, the CN-OA-m photoelectrode delivers a markedly

higher photocurrent, underscoring its superior electron–hole separation and transport. Electrochemical impedance spectroscopy (EIS) supports this observation, the CN-OA-m (Figs. 7G and S24) displays a significantly smaller semicircle radius compared to the other $C_3N_4$ samples, signifying lower interfacial charge-transfer resistance and faster electron mobility. These findings further corroborate the efficient electron transfer from the emissive state to Ni(II), facilitating the generation of Ni(I) species. Time-resolved photoluminescence (TRPL) spectra monitored at the corresponding emission peaks give the average radiative lifetimes (τ) of the recombining charge carriers. The radiative lifetimes of p-$C_3N_4$, g-$C_3N_4$, mpg-$C_3N_4$, $C_3N_4$, PR-$C_3N_4$, MC-$C_3N_4$, CN-OA-m were 1.52, 2.04, 2.48, 2.78, 3.62, 3.73, and 3.88 ns, respectively (Figs. 7H, S25). Among these catalysts, CN-OA-m exhibits the longest fluorescence lifetime, suggesting a reduced recombination rate of photogenerated electron-hole pairs. This extended lifetime implies that photogenerated electrons are more effectively sustained during the reaction, thereby enhancing the overall photocatalytic efficiency. The electrochemical potential diagrams (Fig. 7I) of several catalysts[29,33,55,84–87] further show that CN-OA-m possesses the highest reduction potential, indicating the possibility of a photoredox reaction. Combined with the above analysis, porous CN-OA-m possess the efficient separation efficiency of photogenerated electronhole pairs, exhibits relatively superior properties and may be the best photocatalyst for the reactions examined, benefiting the photocatalytic carbon–heteroatom cross-coupling reaction.

## Synthetic application and mechanistic considerations

Red light penetrates approximately 23 times further into reaction solutions than blue light, reaching a depth of 12 cm before 90% of its power is absorbed by the solution[42]. To highlight the utility of red light-promoted cross-coupling reactions, we performed a gram-scale reaction of aryl bromide and n-butylamine in a round-bottom flask with a diameter of 10 cm, which yielded tetracaine (**263**) in 85% yield (Fig. 8A), further demonstrating the utility of red light's penetration ability in this semiheterogeneous catalysis. Impressively, even when simulated solar light was used as the light source, tetracaine was obtained in 86% yield (Fig. 8A, See SI sections 4.5 and 4.7). Additionally, the yield of tetracaine was 59% when the reaction was carried out under solar light for 5 h (Fig. 8A, See SI section 4.4). Furthermore, a significant advantage of this method is that the photocatalyst CN-OA-m could easily be recycled and reused (Fig. 8B; and see section 5 in SI). After the reaction was complete, the CN-OA-m powder was recovered by centrifugation and then washed. Fourier transform IR spectroscopy (Fig. S29), UV–vis spectroscopy (Fig. S30), scanning electron microscopy (Fig. S31 and S32), transmission electron microscopy (Fig. S33 and S34), and solid-state NMR spectroscopy (Fig. S35) confirmed that the recovered CN-OA-m retained its original structural characteristics. It could be reused five times under identical conditions with only a slight decrease in the yield of the coupling product[77]. Taken together, the post-characterization data for the recovered CN-OA-m clearly demonstrated the robustness, stability, and durability of this semiheterogeneous photocatalyst.

Next, we carried out mechanistic studies. The UV–vis spectrum of the prepared CN-OA-m showed an absorption band in the red-light region (Fig. S16-17). Additionally, after a solution of CN-OA-m in DMAc was irradiated with red light for 10 min, a signal with a g value of 2.003 was observed in the electron paramagnetic resonance spectrum (Fig. 8C; and See SI section 11.2), a finding that is consistent with literature reports[61,65] and that suggests that red-light irradiation produced electrons. Moreover, the progress of the red-light-promoted reaction slowed dramatically after the addition of an electron (e⁻) (AgNO₃ or KI) or a hole (h⁺) scavenger (Na₂S/Na₂SO₃ or EDTA) (Fig. 8D; and See SI section 7)[66,88]. These results confirm that red-light excitation of photogenerated charges played a major role in this photocatalytic coupling reaction. Furthermore, when CN-OA-m (2 mg) was used as the

photocatalyst and a Ni(II)(ligand)Br₂ complex was the metal catalyst under red-light irradiation for 90 min, electron paramagnetic resonance spectroscopy revealed a distinct signal at $g_{avg}$ = 2.204 (Fig. 8E; and See SI section 11.1)[89–91], indicating the formation of a Ni(I) species. Therefore, we speculate that upon red-light excitation, CN-OA-m served as an electron donor to reduce the Ni(II)(ligand)Br₂ complex to a Ni(I) species via a single-electron transfer. In addition, in the absence of substrates, the Ni(II)(ligand)Br₂ complex was irradiated with red light for 90 minutes in the presence of CN-OA-m, aiming to in situ produce the presumed Ni(I) species; and then an aryl bromine and an amine were introduced and allowed to react in the dark for 24 h. This experiment delivered the desired product in 29% yield by NMR spectroscopy (Fig. 8F; and see SI section 10.7). These results indicate that under red-light irradiation, a catalytically active Ni(I) species was generated.

Our previous work[89,92–97] and the above-mentioned observations support the view that CN-OA-m absorbs photons of enough energy under red light irradiation, excitation of electrons from the valence band potential (VB, $E_{VB}$ = 0.88 V) to the conduction band potential (CB, $E_{CB}$ = −1.65 V) occurs with consequent charge separation (formation of a hole-electron pair), the as-formed electrons (e⁻) from the emissive state to reduce Ni(II) ($E_i$ [Ni$^{II}$/Ni$^{I}$] = −1.43 V vs SCE)[98,99] to generate a Ni(I) species via SET, suggesting that this cross-coupling reaction likely proceeds via a Ni(I)/Ni(III) cycle (Fig. 9). Specifically, the absorption of red light by CN-OA-m triggers a charge-separation process that produces redox centers in the form of electron–hole pairs. The electrons in the conduction band in the photogenerated hole reduce Ni(II) via a single-electron transfer to form the active Ni(I) species, and the electron of organic base (mDBU, $E_{p/2}$ = +1.39 V vs Ag/AgCl in MeCN) transfer to the holes[58,59,62,66] to complete the photocatalytic cycle. Concurrently, the Ni(I) species undergoes an oxidative addition reaction with the aryl halide to form a Ni(III) complex, which undergoes ligand exchange with the nucleophile. The resulting species is deprotonated by the organic base, yielding an Ar-Ni(III)-Nu species, which then undergoes a reductive elimination reaction to release the ArNu coupling product and a Ni(I) species to start the next catalytic cycle. When the illumination is stopped, the formation of a catalytically inactive Ni(II) species by comproportionation of Ni(III) and Ni(I) species[89,92–96,98,99] or reoxidation[60] of Ni(I) by the organic base radical cation interrupts the reaction. Continuous illumination is crucial to converting these inactive, off-cycle Ni(II) species back to the active Ni(I) species, thereby maintaining the catalytic cycles.

In summary, we have developed a method for red-light-driven Ni-catalyzed C–heteroatom cross-coupling reactions using CN-OA-m as a photocatalyst. The versatile Ni/red-light semiheterogeneous catalyst system enables the formation of four different types of C–heteroatom bonds and has a broad substrate scope. Notably, the use of a low-energy red light results in broad functional group tolerance, demonstrating the method's excellent potential utility for pharmaceutical chemistry. Catalyst-recycling experiments demonstrated that the CN-OA-m could be recovered and reused at least five times. This method provides a new platform for the development of other reactions involving red-light metallaphotoredox catalysis.

## Methods

### Standard procedure for the C-N couplingh

To an oven-dried 10 mL of storage tube were added NiBr₂·glyme (10.0 mol%), 1 mL of DMAc and a magnetic stir bar under argon atmosphere. The mixture was evacuated and backfilled with argon for 3 times. Then the aryl bromide (0.2 mmol), amine (0.4 mmol), CN-OA-m (10 mg/mL) and mDBU (1,4,5,6-tetrahydro-1,2-dimethylpyrimidine) (1.5 eq., 0.3 mmol) were added. The tube was sealed with a Teflon screw valve. The reaction mixture was then irradiated with 10 W red LEDs (0.5 cm away from the tube, 660–670 nm) at 85 °C. After the reaction was completed, the mixture was diluted with ethyl

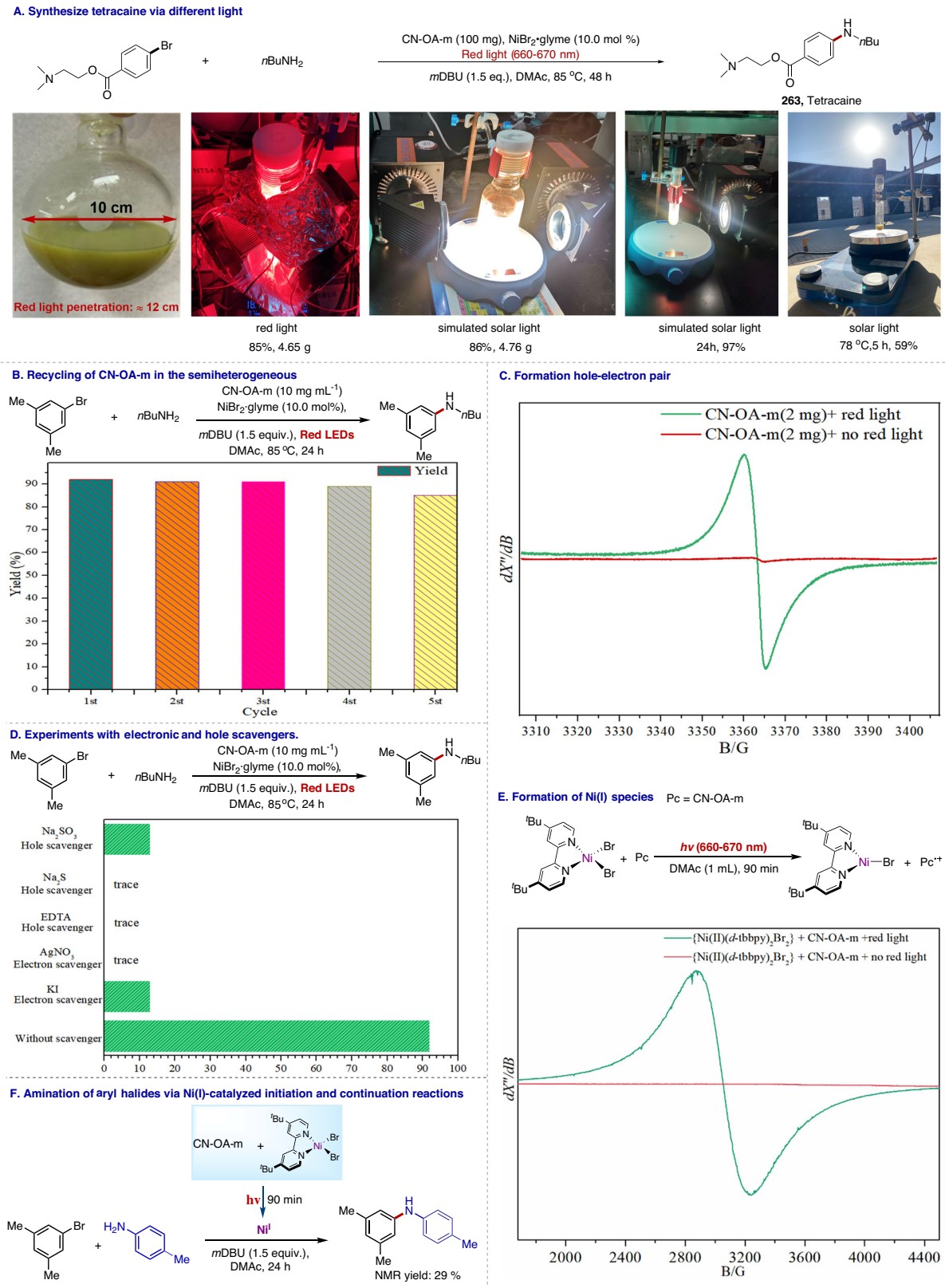

**Fig. 8 | Synthetic application and mechanistic considerations. A** Synthesis of tetracaine via coupling reactions under various light sources; (**B**) Recycling of CN-OA-m after coupling reactions under standard conditions for 24 h; (**C**) Electron paramagnetic spectroscopic detection of a nitrogen vacancy (298 K);

(**D**) Experiments involving electron and hole scavengers; (**E**) Detection of Ni(I) species (100 K); (**F**) Ni(I) initiated catalytic coupling reaction. For details, see Supporting Information.

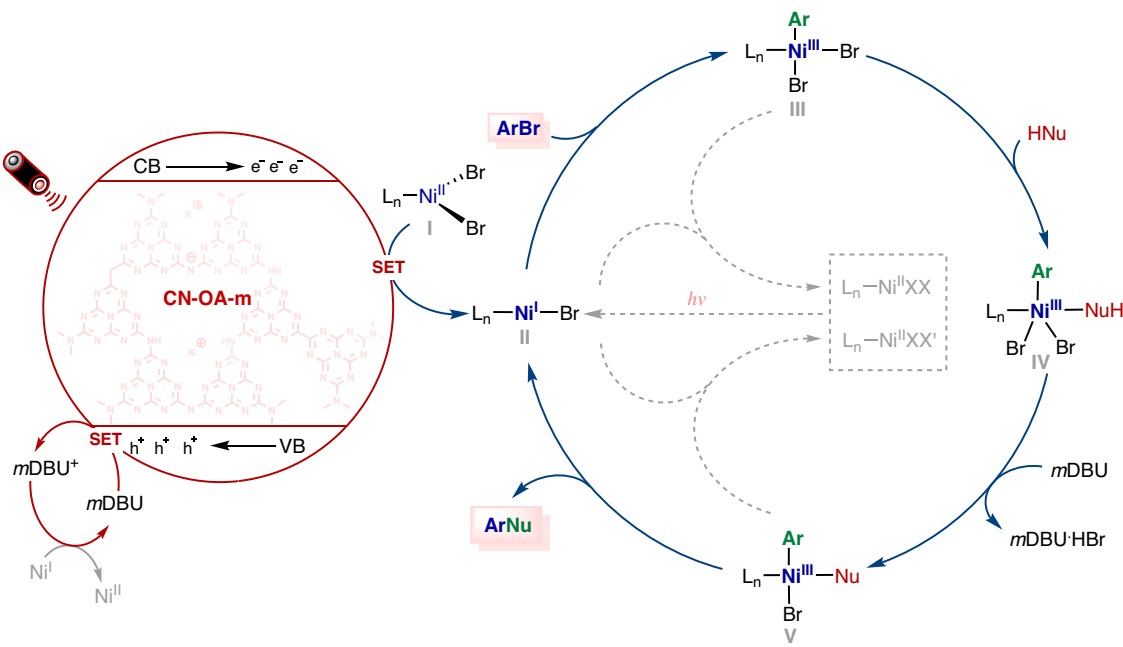

**Fig. 9 | Tentative Mechanism.** Proposed catalytic cycle for the red-light-driven Ni-catalyzed cross-coupling reaction.

acetate after cooling to room temperature. The organic phases were washed with saturated ammonium chloride (3 × 10 mL), dried over anhydrous sodium sulfate, and concentrated under reduced pressure. The residue was purified by flash column chromatography using petroleum ether and ethyl acetate as eluent to afford aryl amines.

### Standard procedure for the C-O coupling

To an oven-dried 10 mL of storage tube were added *d*-tbbpy (4,4'-Di-tert-butyl-2,2'-dipyridyl) (10.0 mol%), NiBr$_2$·glyme (10.0 mol%), 1 mL of DMF and a magnetic stir bar under argon atmosphere. The mixture was evacuated and backfilled with argon for 3 times. Then aryl bromide (0.2 mmol), *O*-nucleophilic reagent (0.6 mmol), CN-OA-m (10 mg/mL) and *m*DBU (1,4,5,6-tetrahydro-1,2-dimethylpyrimidine) (1.5 eq., 0.3 mmol) were added. The tube was sealed with a Teflon screw valve. The reaction mixture was then irradiated with 10 W red LEDs (0.5 cm away from the tube, 620–630 nm) at 85 °C. After the reaction was completed, the mixture was diluted with ethyl acetate after cooling to room temperature. The organic phases were washed with saturated ammonium chloride (3 × 10 mL), dried over anhydrous sodium sulfate, and concentrated under reduced pressure. The residue was purified by flash column chromatography using petroleum ether and ethyl acetate as eluent to afford aryl ethers.

### Standard procedure for the C-S/Se coupling

To an oven-dried 10 mL of storage tube were added *d*-tbbpy (4,4'-Di-tert-butyl-2,2'-dipyridyl) (10.0 mol%), NiBr$_2$·glyme (10.0 mol%), 1 mL of DMAc and a magnetic stir bar under argon atmosphere. The mixture was evacuated and backfilled with argon for 3 times. Then aryl bromide (0.2 mmol), *S*-nucleophilic reagent (0.4 mmol), CN-OA-m (10 mg/mL) and *m*DBU (1,4,5,6-tetrahydro-1,2-dime-thylpyrimidine) (1.5 eq., 0.3 mmol) were added. The tube was sealed with a Teflon screw valve. The reaction mixture was then irradiated with 10 W red LEDs (0.5 cm away from the tube, 620–630 nm) at 85 °C. After the reaction was completed, the mixture was diluted with ethyl acetate after cooling to room temperature. The organic phases were washed with saturated ammonium chloride (3 × 10 mL), dried over anhydrous sodium sulfate, and concentrated under reduced pressure. The residue

was purified by flash column chromatography using petroleum ether and ethyl acetate as eluent to afford aryl sulfides.

## Data availability

Data are available in this study are provided in the Supplementary Information. All other data are available from the authors upon request.

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

## Acknowledgements

This research is supported by the National Natural Science Foundation of China (Grant No. 22171174, 22471150 to D.X.; Grant No. 22402113 to G.S., Grant No. 21872089 to Q.G.), the Fundamental Research Funds for the Central Universities (Grant No. GK202406026 to T.K.), the Innovation Capability Support Program of Shaanxi (Grant No. 2023-CX-TD-28 to D.X.), the Fundamental Science Research Project of Shaanxi for Chemistry, Biology (Grant No. 22JHZ002 to D.X.), the Natural Science Foundation of Shaanxi Province (Grant No. 2019JM-043 to D.X., Grant No. 2024JC-YBQN-0075 to G.S., Grant No. 2025JC-YBQN-144 to T.K., Grant No. 2025JC-YBMS-148 to Q.G.), and the State Key Laboratory of Natural and Biomimetic Drugs (Grant No. K202437 to to G.S.).

## Author contributions

D.X. conceived and directed the project. G.S. synthesized catalysts, designed and conducted experiments, analyzed data. W.Z. synthesized various $C_3N_4$ catalysts, conducted their characterization and discussed their catalytic activity. J.S. and Q.L. participated the substrate scope expansion. Y.F., J.F., H.L. helped with catalyst synthesis and characterization. T.K., J.D., G.L., X.P.Z., C.W. contributed to project discussion. Q.G. participated in the guidance of synthetic catalysts and revised the manuscript. G.S. prepared the manuscript. D.X. finally wrote the manuscript. All authors discussed the experimental results and commented on the manuscript.

## Competing interests

The authors declare no competing interests.
