## [Transparent Peer Review file · Nature Communications]

General Method for Carbon–Heteroatom Cross-Coupling Reactions via Semiheterogeneous Red-Light Metallaphotocatalysis

Corresponding Author: Professor Dong Xue

Version 0:

Reviewer comments:

Reviewer #1

(Remarks to the Author)

Photocatalysis has been at the forefront of research for several decades, including various systems. In most of its manifestations, UV and photons on the short-wavelength edge of the electromagnetic spectrum have been used to excite the photocatalyst. The preference for photons of higher energy is dictated by a few factors:

1. The photophysics and energy losses inherent to a majority of common photocatalysts – the formation of a photocatalyst long-lived triplet state proceeds via a singlet state of higher energy.

2) The difference between the photocatalyst excited state reduction and oxidation potentials must staddle redox potentials of substrates, i.e., the photocatalyst excited state must be able to oxidize/reduce thermodynamically stable substrates, which possess strongly positive oxidation potential or strongly negative reduction potential, respectively.

This work reports application of a combination of CN-OA-m, a type of graphitic carbon nitride, and Ni-salts as transition metal catalyst to drive various cross-coupling reactions upon irradiation with red light. Overall, the results are interesting. Given the abovementioned reasons of dominance of using more energetic photons in photocatalysis, the results are also important and new. The article may become suitable for publication in Nature Communications after authors strengthen their manuscript with structure-activity relationship study, the role of temperature in the studied reaction, mechanism of Ni(I) species formation, and others.

1. In the abstract “semihomogeneous catalyst” is mentioned, while in the manuscript “semiheterogeneous catalysis” is encountered several times. What concepts do authors investigate here?

2. Section “Results”. Energy is expressed, among other units, in electron-volts (eV), while potential is in volts (V). Please correct.

3. A study of structure-activity relationship should be conducted to identify a property/structural feature (specific surface area, surface chemical structure, photophysics of the bulk, etc.) of CN-OA-m that makes this type of carbon nitride the most active in the studied reaction.

4. Temperature seems to be a critical element that enables the reaction under red light. Perhaps, it is even more important than the employed photon wavelength. From the study of reaction yield at different temperatures, authors could determine activation energy and compare it with existing photocatalytic/non-photocatalytic processes.

5. Throughout the manuscript authors refer to aryl bromides as “aryl halides”. Aryl bromides, especially electron deficient, are more reactive compared to electron-rich aryl bromides and aryl chlorides. As such, authors invested substantial time into this not very challenging reaction. Is the developed photocatalytic system also suitable to couple aryl chlorides with various nucleophiles?

6. Section “Synthetic Application and Mechanistic Considerations”. Authors claim that the depth of red light penetration is 12 cm. How was this number obtained? Are there any experimental evidences collected in this project to support this claim?

Also, authors need to verify if reaction efficacy is really due to better light delivery into the bulk of the reaction mixture compared to using near-UV/UV photons.

7. Generation and using Ni(I) species. Reduction of Ni(II) complex into Ni(I) implies that some part of the complex or other components of the reaction mixture is oxidized? What are these molecules/reductants and what are the products of their oxidation? Please identify the products and suggest a reaction equation that can describe this process of Ni(I) formation?

8. Figure 1C,E legend. It must be indicated that the spectra were acquired at 100K.

9. Figure 1C title and reaction scheme. What is “nitrogen hole”?

10. In Figure C,E, the scale of X-axis should be the same to emphasize on different EPR signals widths and their position.

Also, do Ni(I) species form in the absence of CN-OA-m? Do they form upon irradiation with UV/near-UV photons in the absence of CN-OA-m?

11. While the manuscript writing style is overall sufficiently clear, there are grammatical errors and unusual expressions, mainly in figures and schemes. They must be corrected. Here are some of them:

11.1. Scheme 1, "reactions is UERgent".

11.2. Scheme 1, consider to rephrase a fragment of the text as "Is red-light metallaphotocatalytic C-heteroatom cross-coupling possible?"

11.3. Figure 1D title. Consider to revise as "Experiments with electron and hole scavengers".

11.4. Figure 1A, "simulated solar light".

11.5. EPR spectra. What is dX/dB ? Second derivative of X with respect to magnetic field strength?

11.6. "suggests that red-light irradiation produced electrons." Irradiation with light triggers electron transition between the valence and the conduction band, and formation of a hole-electron pair. A photogenerated electron/hole may be imagined as an open-shell species localized on carbon/nitrogen atom, respectively. As such, irradiation with light does not "produce electrons", but creates an electronically excited species, which is a biradical (singlet or triplet in nature).

11.7. "electron (e-) sacrifier (AgNO₃ or KI) or a hole (h+) sacrifier". Scavenger in both instances.

11.8. Typos in authors' names and papers' titles in references. In particular, in reference 1, 43, 44, 49, 53, 54, 67, 68, 71, 72, 76, 78, 81, 94, etc.

Reviewer #2

(Remarks to the Author)

In the manuscript entitled "General Method for Carbon-Heteroatom Cross-Coupling Reactions via Semiheterogeneous Red-Light Metallaphotocatalysis" the authors applied heterogeneous polymeric carbon nitride (CN-OA-m) under red light illumination for driving the nickel-catalyzed cross-coupling reactions. This catalytic system enabled the formation of four different types of carbon-heteroatom bonds (C-N, C-O, C-S, and C-Se) with a wide range of substrates (more than 200 examples). This study explores an interesting area of heterogeneous photocatalysis for cross-coupling reactions, although there are several previous literatures reported the cross-coupling reactions by using carbon nitride. However, the manuscript lacks a comprehensive explanation regarding why the material exhibits optimal activity under red light than blue light, yet shows no activity when exposed solely to red light, requiring the combination of red light and heating for activation. Additionally, the observed high yields (>50%) in several other carbon nitride materials such as C₃N₄ and mpg-CN that do not respond to red light further complicate the assertion that the reaction is truly driven by red light. As a result, I find it difficult to fully accept that red light is the primary driving force behind the reaction. Consequently, the proposed reaction mechanism under red light requires further experimental validation and supporting data. In its current form, the manuscript should not be considered for publication in Nature Communications.

1. In Table 1, materials such as C₃N₄, mpg-CN, p-C₃N₄, and g-C₃N₄ are generally considered to lack infrared responsiveness. However, the high activity observed under red light in this study, with yields consistently exceeding 50%, is puzzling. What accounts for this phenomenon? Based on these results, I am inclined to question the assertion that the reaction is driven by red light.

2. In Table 1, CN-OA-m, when applied to photocatalytic HER (refer to: *Angew. Chem. Int. Ed.* 56, 13445 (2017)), shows significantly higher activity under blue light compared to green and red light. Why, then, does the activity under red light appear to be lower in this case? What is the underlying cause?

3. When using CN-OA-m under blue light source, but without heating, what is the observed yield?

4. Previous literature reports that carbon nitrides such as mpg-CN and CN-V, when coupled with Ni catalysts under blue light, can successfully drive C-N coupling reactions (refer to *Science* 365, 360 (2019) and *Cell Rep. Phys. Sci.* 2, 100491 (2021)). Although red light is employed in your study, heating is also required. How does this approach demonstrate an advanced or novel aspect of your work?

5. If only red light is used for excitation without heating, resulting in no activity, can we conclude that the CN-OA-m does not actually respond to red light? Does this imply that the system should not be classified as a photocatalytic system? Is the red light here primarily acting in a thermal capacity, but possibly differing from direct heating? What is the specific mechanism behind this behavior? This uncertainty in the reaction mechanism requires further experimental validation to confirm whether the reaction is truly driven by red light. This remains a significant issue in the manuscript.

6. In Table S6, the trends observed for different light sources are quite unusual. The catalytic effect of green LED (520 nm–530 nm) is higher than that of the 580–595 nm light source, yet when using a 620 nm–630 nm source, the yield exceeds that of the green LED. What explains this unexpected trend?

7. Based on the DRS and Figure S25, it is apparent that Ni particles are deposited on CN-OA-m. Furthermore, the relative ratio of the two peaks at 166 and 159 in the NMR spectrum shows a significant change. Shouldn't this indicate substantial structural alterations? Therefore, the claim of minimal structural change seems inconsistent. What caused this structural change observed in the NMR spectra?

8. The captions for Figures S21–S27 in the Supplementary Materials are all incorrect.

9. What are the conduction band and valence band positions of CN-OA-m? Does it possess sufficient reduction potential to

reduce NiII/Ni (-1.43 V vs SCE)?

10. In Figure 1C, the EPR results maybe inaccurate. According to previous studies using EPR to characterize carbon nitrides, a distinct peak for unpaired electrons around $g=2.0$ should be observed under dark conditions even with standalone carbon nitride (see *Angew. Chem. Int. Ed.*, 2019, 58(11): 3433-3437). However, this peak is absent here, which is inconsistent. This cannot serve as evidence for infrared responsiveness.

11. In Figure 1E, the EPR results are also problematic. Why is there no signal near $g=2.0$ for CN-OA-m? Additionally, EPR peaks for Ni(I) typically show splitting, yet this is not observed here, which is unreasonable (see *Angew. Chem. Int. Ed.*, 2023, 62(43): e202310753).

Reviewer #3

(Remarks to the Author)

Compared to molecular transformation reactions using blue-to-green light, red-to-near-infrared (NIR) light has clear advantages, but such research has been limited. A key challenge has been the development of catalysts capable of effectively interacting with red-to-NIR light while exhibiting appropriate reactivity with substrates. The authors propose carbon nitride-type heterogeneous semiconductors as a novel class of red-light photocatalysts to address this issue. Specifically, the authors focus on the recently reported CN-OA-m catalyst and its application to red-light-driven nickel-catalyzed carbon-heteroatom cross-coupling reactions.

One of the primary advantages of this reaction is its broad applicability to a wide range of nucleophiles without case-by-case optimization of the reaction conditions. Additionally, it leverages the benefits of heterogeneous catalysis, demonstrating recyclability. The reaction mechanism is thoroughly analyzed with appropriate references to previous findings.

While these findings underscore the potential of CN-OA-m as a novel red-light photocatalyst, the advantages of utilizing red light were not sufficiently addressed. My concerns are outlined below. If these aspects are adequately clarified, this paper will be one of the standards in the chemistry of red-light-driven photocatalysts.

1. General reaction conditions applied to various substituents and nucleophiles represent a significant advantage for this reaction. This reaction proceeds via two catalytic cycles as the authors described (Scheme 5): one involving CN-OA-m photocatalyst and red light and another involving a nickel catalyst. While a broad range of nucleophiles can be used, their reactivity is governed primarily by the latter catalytic cycle. Since this cycle does not require light, the observed advantage may not be unique to the CN-OA-m photocatalyst. Could some nickel complex intermediates in Scheme 5 absorb shorter-wavelength light and compete with undesired reactions? If this hypothesis is valid, can the authors provide spectroscopic data on the intermediates to support it?

2. The authors described that this reaction requires light and heat (85°C). If the proposed reaction mechanism is correct, heat is essential for the nickel-catalyzed process. The nickel-catalyzed process should have been previously reported, and a more detailed discussion of the role of heat is necessary.

3. What temperature were the measurements in Figure 1E conducted? Can the effect of heat be further discussed through temperature-dependent measurements?

4. While the recycled catalysts have been characterized using various measurements, can the authors also include XRD analysis?

5. The authors proposed that mDBU is the optimized base to react with the hole of the catalyst. Is the oxidation potential of mDBU consistent with this role? Can the authors discuss the differences in oxidation potentials of bases presented in Table S7?

6. The broad scope of this reaction to a wide range of substrates and nucleophiles is highly valuable. However, most of the substrates can interact with only UV light. One of the key advantages of red-light photocatalysis, as noted by the authors, is that it minimizes competition with the absorption of substrates. Therefore, I strongly recommend the authors demonstrate the applicability of this reaction to substrates that strongly absorb visible light, such as highly π -conjugated molecules.

7. The reactions presented in this manuscript settle on one reaction mechanism. Further exploration of CN-OA-m's applicability to other photochemical reactions should be valuable, while many examples are not required. Such results can provide crucial insights into whether conventional photochemical reactions can be adapted for red-light-driven reactions using this catalyst.

Version 1:

Reviewer comments:

Reviewer #1

(Remarks to the Author)

In this revised version of the manuscript, authors addressed satisfactory ~80% of my original comments. Comments from the original report that require further improvements are:

1. Please report specific surface areas of carbon nitrides that were obtained from N₂ physisorption measurements.

2. Section "Results and discussion", first sentence "conduction band"  "conduction band potential" and "valence band"  "valence band potential". Also, please indicate the reference electrode.

3. Based on group's earlier publication and considering redox potentials, authors assumed that mDBU is oxidized. Attempts to detect and/or isolate any products of mDBU oxidation were not made.

In addition, a few questions arose after reading the revised manuscript:

1. Authors measured oxidation potential of bases. In particular, they found that oxidation potential of mDBU (presumably peak) is +2.14 V vs. Ag/AgCl. Given the potential of the valence band of CN-OA-m of +0.8 V indicated in the Results and Discussion section, electron transfer from this reactant to CN-OA-m electronically excited state must be strongly endergonic, and as such does not proceed at significant rate. More correct approach, which is actually adopted in literature, is to report (Ep/2) potential at half-peak. See for example, Synlett 2016; 27(05): 714-723. This will give a more reasonable value.

2. Discussion of the mechanism, "the as-formed holes (h+) from the emissive state to reduce Ni(II) (Ei [NiII/NiI] = -1.43 V vs SCE) to generate a Ni(I) species via SET". Probably authors mean "the as-formed electrons (e-)".

3. Authors corrected some typos, but introduced other new typos.

3.1. Figure 2C. X-axis label "maNgnetic field". Y-axis label "eRP siNGal". Also, is there any good reason why authors use different axes labels in Figure 2C and 2E, "B/G" vs. "maNgnetic field" and "dX"/dB" vs. "eRP siNGal"?

3.2. Scheme 1, CN-OA-m structure. It appears that one of the bridging N-atom is methylated. Is it a typo?

3.3. Section title "C-heteroatom couplings".

Overall, the scientific content is of high quality. However, the quality of writing requires substantial improvement. I will leave this aspect to the authors and editor's discretion. If it is acceptable to publish a paper with obvious typos in a flagship journal of the Nature family journals

Reviewer #2

(Remarks to the Author)

REVIEW 2# — Joint Evaluation with Reviewer 3# Comments

Thank you for inviting me to re-evaluate the revised manuscript entitled "General Method for Carbon-Heteroatom Cross-Coupling Reactions via Semiheterogeneous Red-Light Metallaphotocatalysis" by Song et al.

In this resubmission, the authors have made several substantial improvements. Notably, they addressed critical points regarding the advantages of CN-OA-m, broadened the substrate scope, investigated the role of reaction temperature, and incorporated additional characterizations to support their structure-activity relationship analysis. These revisions, along with the correction of previous inaccuracies, have resulted in a clearly enhanced manuscript. This present article may become suitable for publication in Nature Communications. While, several concerns still require clarification:

1. In Figure S20, the N₂ adsorption/desorption isotherms for MC-C₃N₄ and RP-C₃N₄ are not closed, which is abnormal and suggests potential issues with the measurements. Re-measurement is strongly recommended.

2. In Figure S20, the light absorption of p-C₃N₄, g-C₃N₄, and C₃N₄ beyond 550 nm is extremely weak. It is difficult to believe that such minimal absorption would lead to any significant photoactivity under red-light irradiation. Moreover, the DRS spectra should include labeled y-axes. It is advised that the authors provide more convincing evidence demonstrating that these materials indeed exhibit red-light-responsive photocatalytic behavior under specific thermal conditions.

3. Considering the thermal radiation associated with red-light illumination, it is recommended that the surface temperatures of the catalyst be measured separately under (i) red-light irradiation alone and (ii) combined red-light and thermal heating conditions.

4. In Figure S36, a new XRD peak appears at approximately 40° in the recycled catalyst. What is the origin of this peak? Further explanation or identification is required.

Reviewer #4

(Remarks to the Author)

For the DFT calculations, the methods and basis sets of geometry optimization and single point energy calculation are acceptable. The solvation effect is also considered to ensure the credibility of the results. However, I disagree with the statement "the activation energy requisite

for the oxidative addition of electron-rich aryl bromides amounts to 41.5 kJ/mol, indicating it

is quite endothermic. This further suggests that temperature is crucial in oxidative addition involving Ni(I) species". It is generally believed that reactions with free energy barriers lower than 20 kcal/mol can spontaneously occur at room temperature, For example K. N. Houk* Journal of the American Chemical Society 2022 144 (4), 1971-1985.

Version 2:

Reviewer comments:

Reviewer #1

(Remarks to the Author)

Authors addressed all the remaining issues. The manuscript may, in principle, be recommended for publication in Nature Communications.

Reviewer #2

(Remarks to the Author)

The issues mentioned have already been addressed.

Responses to the Comments by Reviewer: 1

General Comment:

Photocatalysis has been at the forefront of research for several decades, including various systems. In most of its manifestations, UV and photons on the short-wavelength edge of the electromagnetic spectrum have been used to excite the photocatalyst. The preference for photons of higher energy is dictated by a few factors: 1) The photophysics and energy losses inherent to a majority of common photocatalysts—the formation of a photocatalyst long-lived triplet state proceeds via a singlet state of higher energy; 2) The difference between the photocatalyst excited state reduction and oxidation potentials must staddle redox potentials of substrates, i.e., the photocatalyst excited state must be able to oxidize/reduce thermodynamically stable substrates, which possess strongly positive oxidation potential or strongly negative reduction potential, respectively; 3) This work reports application of a combination of CN-OA-m, a type of graphitic carbon nitride, and Ni-salts as transition metal catalyst to drive various cross-coupling reactions upon irradiation with red light. Overall, the results are interesting. Given the abovementioned reasons of dominance of using more energetic photons in photocatalysis, the results are also important and new. The article may become suitable for publication in Nature Communications after authors strengthen their manuscript with structure-activity relationship study, the role of temperature in the studied reaction, mechanism of Ni(I) species formation, and others.

Response: We thank the reviewer's valuable advice. The manuscript has been carefully revised based on your comments and suggestions.

Specific Comment

1. In the abstract "semihomogeneous catalyst" is mentioned, while in the manuscript "semiheterogeneous catalysis" is encountered several times. What concepts do authors investigate here?

Response: We thank the reviewer's valuable advice. Due to our negligence, we mistakenly wrote "semiheterogeneous catalysis" as "semihomogeneous catalyst." This error not only affects the accurate expression of the terminology, but may also mislead readers, this has now been corrected to semiheterogeneous catalysis in the manuscript.

2. Section “Results”. Energy is expressed, among other units, in electron-volts (eV), while potential is in volts (V). Please correct.

Response: We thank the reviewer’s valuable advice. We have corrected the relevant descriptions in the manuscript.

3. A study of structure-activity relationship should be conducted to identify a property/structural feature (specific surface area, surface chemical structure, photophysics of the bulk, etc.) of CN-OA-m that makes this type of carbon nitride the most active in the studied reaction.

Response: We really appreciate the reviewer’s valuable advice. According to the reviewer’s suggestion, we have investigated several of the catalysts used in this paper (C₃N₄, p-C₃N₄, g-C₃N₄, mpg-C₃N₄, RP-C₃N₄, MC-C₃N₄, CN-OA-m) for their property and structural feature, including, XPS spectra, specific surface area, photoluminescence (PL), photocurrent responses, electrochemical impedance spectra, Time-resolved photoluminescence (TRPL) spectra, band structures (see below):

(1) X-ray photoelectron spectroscopy (XPS) analysis of CN-OA-m indicates that its surface elemental composition comprises C, N, and K elements (please also refer to Figure S18 in SI). The analysis of the N 1s spectrum for CN-OA-m reveals three peaks at 399.2, 398.4, and 397.7 eV, which can be attributed to C–NH_x, N–C₃, and C–N=C, respectively. The additional weak peak of N1s at 396.7 eV is attributed to the negatively charged C–N=C group, which can be neutralize the positive K⁺. Furthermore, the C 1s spectrum of CN-OA-m shows peaks at 288.0, 286.0, and 284.7 eV, which are attributed to N–C=N, C–NH_x, C–C. Those findings further suggest that defect energy levels can be generated between the conduction band and the valence band, and greatly reduce the recombination of photogenerated electron-hole pairs in CN-OA-m by trapping photogenerated electrons, which is beneficial to the enhancement of the photoredox process (ref: *Angew. Chem. Int. Ed.* **2018**, *57*, 10246; *Adv. Mater.* **2017**, *29*, 1605148).

(2) The surface area and porosities of those catalysts, N₂ adsorption at 77 K was performed. The N₂ adsorption isotherm of all semiheterogeneous photocatalysts, the CN-OA-m shows a typical type IV curve with a classic H3-type hysteresis loop, indicating that the material has an abundant porous structure (please also refer to Figure S20). The Brunauer–Emmett–Teller (BET) surface areas calculated for those catalyst, the CN-OA-m are 43.8 cm³g⁻¹ with average pore diameters of 13.80 nm.

The BET pore size distribution of CN-OA-m exhibits the micro-meso-macropores characteristics (please also refer to Figure S21), with a particularly significant proportion of macropores compared to other semiheterogeneous photocatalysts. The abundant porous structure of CN-OA-m provides a large surface area to fully expose the active sites and facilitates the diffusion pathways of photogenerated charge carriers enhanced photocatalytic activity. (See below, and please also refer to SI Figure S20, 21) (ref: *Mater. Horiz.* **2017**, *4*, 493–501).

(3) Among these different semiheterogeneous photocatalysts, p-C₃N₄, g-C₃N₄, mpg-C₃N₄ demonstrate notably strong photoluminescence (PL) intensity, indicating severe recombination of photogenerated charge carriers. In contrast, the PL intensity of C₃N₄, RP-C₃N₄, MC-C₃N₄, CN-OA-m exhibits a sequential decrease, implying suppressed recombination of charge carriers. This trend is most pronounced in CN-OA-m, which shows the weakest PL intensity, suggesting an efficient electron transfer from the emissive state to Ni(II) species. This process likely facilitates the formation of Ni(I) species (See below, and please also refer to SI Figure S22) (ref: *Angew. Chem. Int. Ed.* **2017**, *56*, 13445; *Nat Commun.* **2023**, *14*, 1501).

(4) Time-resolved photoluminescence monitored at the corresponding emission peaks give the mean radiative lifetimes (τ) of the recombining charge carriers. The τ of p-C₃N₄, g-C₃N₄, mpg-C₃N₄, C₃N₄, RP-C₃N₄, MC-C₃N₄, CN-OA-m were 1.52, 2.04, 2.48, 2.78, 3.62, 3.73, and 3.88 ns, respectively (for lifetime components see below, please also refer to SI Figure S23, Table S14). The elongation of fluorescence lifetime indeed signifies a decrease in the recombination rate of photogenerated electron-hole pairs, thereby affording photogenerated carriers an extended duration to engage in photocatalytic reactions. This is beneficial for photocatalytic processes, as highlighted by the notably prolonged residual fluorescence lifetime observed in the CN-OA-m. This extended lifetime implies that photogenerated carriers are effectively sustained throughout the reaction, consequently augmenting the efficiency of the photocatalytic reaction (ref: *Nat Commun.* **2023**, *14*, 1501).

(5) The transient photocurrent response of different C₃N₄ electrodes were measured to gain in-depth insight into charge transport (See below, please also refer to SI Figure S24). Compared with other C₃N₄, the CN-OA-m photoelectrode has a strong photocurrent response, indicating the improved separation and transfer of electrons and holes. Similar results were obtained in electrochemical impedance spectroscopy (EIS) measurements for CN-OA-m (See below, please also refer to the SI Figure S25). The arc radius of the CN-OA-m is significantly smaller than that of other C₃N₄, signifying a lower

resistance to interfacial charge transfer for the enhanced diffusion mobility of electrons from the CN-OA-m, indicative an efficient electron transfer from the emissive state to Ni(II), resulting the Ni(I) species (*Angew. Chem. Int. Ed.* **2017**, *56*, 13445; *Nat Commun.* **2023**, *14*, 1501).

(6) In addition, the electrochemical potential diagrams of several catalysts indicate that CN-OA-m has the highest reduction potential (refer to SI Figure S26) and exhibits high photogenerated electron-hole pair separation efficiency, which facilitates the photocatalytic coupling reaction. This is why it serves as the best photocatalyst for the reaction examined in this study. The corresponding studies and discussions have been supplemented in the manuscript.

Figure S18. XPS spectra of CN-OA-m of C1s, N1s, K2p.

Figure S20. N_2 adsorption/desorption isotherm of different C_3N_4

Figure S21. Pore size distribution of different C_3N_4 .

Figure S22. The photoluminescence (PL) spectra of different C_3N_4 .

Figure S23. Photocurrent response of different C_3N_4 .

Figure S24. The electrochemical impedance spectra (EIS) of different C_3N_4 electrodes.

Figure S25. Time-resolved photoluminescence (TRPL) spectra of different C_3N_4 .

Table S14. Parameters of fluorescence decay curves for C₃N₄-OA-m, mpg-C₃N₄, MC-C₃N₄, RP-C₃N₄, C₃N₄, p-C₃N₄, g-C₃N₄ samples.

Sample	τ_1 (ns) (A ₁ %)	τ_2 (ns) (A ₂ %)	τ_{av} (ns)	Ex (nm)	Em (nm)
CN-OA-m	2.81(97.86)	13.87 (2.14)	3.88	340	533
mpg-C ₃ N ₄	1.86 (98.26)	9.43 (1.74)	2.48	340	457
MC-C ₃ N ₄	2.42(93.81)	9.03 (6.19)	3.73	340	440
RP-C ₃ N ₄	2.33(93.87)	8.82 (6.13)	3.62	340	455
C ₃ N ₄	2.15(98.40)	10.70(1.60)	2.79	340	471
p-C ₃ N ₄	1.19(97.93)	5.11 (2.07)	1.52	340	474
g-C ₃ N ₄	1.43(95.63)	5.49 (4.37)	2.04	340	469

$$\tau_{av} = \frac{A_1\tau_1^2 + A_2\tau_2^2}{A_1\tau_1 + A_2\tau_2}$$

4. Temperature seems to be a critical element that enables the reaction under red light. Perhaps, it is even more important than the employed photon wavelength. From the study of reaction yield at different temperatures, authors could determine activation energy and compare it with existing photocatalytic/non-photocatalytic processes.

Response: We thank the reviewer's valuable advice. According to the reviewer's suggestion, we had studied the effect of different temperatures on reaction yield and found that temperature is indeed critically important for the reaction to proceed. At temperatures below 40 °C, the reaction hardly proceeds. As the temperature rises, the yield increases significantly, but above 90°C, further increases in temperature have a negligible effect on yield (see below, A, also refer to SI section 6). In our early research, we also found that heating has a significant impact on the oxidative addition of Ni(I) species to aryl halides (ref: *J. Org. Chem.* **2022**, 87, 10285; *Chem. Eur. J.*, **2023**, 29, e202300458). Furthermore, through density functional theory (DFT), we found that the activation energy required for oxidative addition of electron-rich aryl bromides is 41.5 kJ/mol (see below, B), further indicating that temperature plays a crucial role in oxidative addition involving Ni(I) species. In addition, by analyzing the changes in yield with temperature, we calculated that the activation energy for the entire reaction process is 51.99 kJ/mol. Through density functional theory calculations and comparison with the literature (ref: *J. Am. Chem. Soc.* **2019**, 141, 6392), this oxidative addition is quite exothermic in the

literature. We have added the corresponding discussion in the manuscript.

A. Effect of temperature on reaction under red light excitation

B. DFT calculation

5. Throughout the manuscript authors refer to aryl bromides as “aryl halides”. Aryl bromides, especially electron deficient, are more reactive compared to electron-rich aryl bromides and aryl chlorides. As such, authors invested substantial time into this not very challenging reaction. Is the developed photocatalytic system also suitable to couple aryl chlorides with various nucleophiles?

Response: We thank the reviewer’s valuable advice. According to the reviewer’s suggestion, we

investigated the coupling of aryl chlorides with various nucleophiles (see below). Fortunately, most aryl chlorides and nucleophiles yielded the corresponding coupling products. We have added the relevant discussion into the manuscript.

Scheme 4. ^a Reaction conditions, unless otherwise noted: aryl chloride (0.2 mmol), NuH (0.4 mmol), CN-OA-m (10 mg mL⁻¹), NiBr₂·glyme (10.0 mol %), DMAc (1 mL), *m*DBU (1.5 equiv.), red light (660-670 nm), 85 °C, Ar, 36 h. ^b 48h. ^c *d*-tbbpy (10.0 mol%), ^d 620-630 nm, NuH (0.6 mmol). Isolated yields are provided. For details, see Supporting Information.

6. Section “Synthetic Application and Mechanistic Considerations”. Authors claim that the depth of red light penetration is 12 cm. How was this number obtained? Are there any experimental evidences

collected in this project to support this claim? Also, authors need to verify if reaction efficacy is really due to better light delivery into the bulk of the reaction mixture compared to using near-UV/UV photons.

Response: We thank the reviewer's valuable advice. The penetration depth of red light is approximately 12.5 cm, as determined through experiments and comparisons with relevant literature (ref. *ACS Cent. Sci.* **2020**, 6, 2053). We utilized red light at 660 nm (intensity 450 mW/cm²), $\epsilon = 399 \text{ M}^{-1}\text{cm}^{-1}$, $c = 2 \times 10^{-4} \text{ M}$. This was calculated using the Beer-Lambert-Bouguer law ($d = 1/\epsilon c$). Additionally, literature reports indicate that the penetration depth of red light is 12 cm (ref. *ACS Cent. Sci.* **2020**, 6, 2053). Furthermore, when we conducted reactions under the same conditions using purple light (390-395 nm), the reaction yield significantly decreased, with an isolated yield of 23% (See below), further demonstrating the excellent penetration capability of red light, where nearly 90% of the photon energy from the red light is absorbed by the reaction mixture (ref. *ACS Cent. Sci.* **2020**, 6, 2053).

7. Generation and using Ni(I) species. Reduction of Ni(II) complex into Ni(I) implies that some part of the complex or other components of the reaction mixture is oxidized? What are these molecules/reductants and what are the products of their oxidation? Please identify the products and

suggest a reaction equation that can describe this process of Ni(I) formation?

Response: We thank the reviewer's valuable advice. When the photocatalyst CN-OA-m is excited by red light, electrons from the valence band transition to the conduction band, forming electron-hole pairs. This is the basis of photocatalysis, as the highly reactive species generated can drive subsequent reactions. The electrons in the conduction band reduce Ni(II) complexes to Ni(I) species through single-electron transfer, while the holes in the valence band oxidize the added reducing agent, an organic base, restoring the photocatalyst back to CN-OA-m and completing the catalytic cycle. Please see the suggested reaction equations for the process of Ni(I) formation listed below.

8. Figure 1C, E legend. It must be indicated that the spectra were acquired at 100K.

Response: We thank the reviewer's valuable advice. Figure 2 C and E were tested at different temperatures. Figure 2C was tested at room temperature (298 K), while Figure 2E was tested at 100 K for metal valence detection. To avoid any misunderstanding for the readers, we have clearly specified the testing temperatures in the manuscript.

9. Figure 1C title and reaction scheme. What is "nitrogen hole"?

Response: We thank the reviewer's valuable advice. To avoid any misunderstanding for the readers, we have revised and corrected the relevant errors and changed nitrogen hole to hole-electron pair in the manuscript.

10. In Figure C, E, the scale of X-axis should be the same to emphasize on different EPR signals widths and their position. Also, do Ni(I) species form in the absence of CN-OA-m? Do they form upon irradiation with UV/near-UV photons in the absence of CN-OA-m?

Response: We thank the reviewer's valuable advice.

i) Figures 2 C and E are tested under different conditions. Figure 2C shows hole testing (free radicals) conducted at room temperature (298K), while Figure 2 E shows the detection of Ni(I) species (metal valence) at 100K. The inconsistency in linewidth makes it difficult to compare the different EPR signal widths and their positions in the same figure;

ii) We have verified through multiple EPR experiments that Ni(I) species do not form in the absence of CN-OA-m. Therefore, under red light conditions, CN-OA-m acts as a photocatalyst providing an electron to reduce Ni(II) to Ni(I) species;

iii) In our Ni chemistry research, we found that Ni(I) species can be formed under near-ultraviolet photon irradiation at 390-395nm (ref: *Chem. Eur. J.*, **2023**, *29*, e202300458; see below).

11. While the manuscript writing style is overall sufficiently clear, there are grammatical errors and unusual expressions, mainly in figures and schemes. They must be corrected. Here are some of them:

11.1. Scheme 1, “reactions is UERgent”.

Response: We thank the reviewer’s valuable advice. Based on your suggestion, this has now been corrected to “urgent” in Scheme 1.

11.2. Scheme 1, consider to rephrase a fragment of the text as "Is red-light metallaphotocatalytic C-heteroatom cross-coupling possible?"

Response: We thank the reviewer’s valuable suggestion. We have already carried out the

substitution with "Is red-light metallaphotocatalytic C-heteroatom cross-coupling possible?" in Scheme 1.

11.3. Figure 1D title. Consider to revise as "Experiments with electron and hole scavengers".

Response: We thank the reviewer's kind suggestion. We have revised the relevant descriptions in Figure 2D.

11.4. Figure 1A, "simulated solar light".

Response: We thank the reviewer's kind suggestion. We have revised the relevant descriptions in Figure 1A.

11.5. EPR spectra. What is dX''/dB ? Second derivative of X with respect to magnetic field strength?

Response: In the context of Electron Paramagnetic Resonance (EPR) spectroscopy, X'' typically refers to the imaginary part of the complex magnetic susceptibility X , which is a function of the applied magnetic field B . The complex magnetic susceptibility X is often written as:

$$X = X' + iX''$$

where:

X' is the real part of the susceptibility, related to the dispersion of the magnetic response.

X'' is the imaginary part of the susceptibility, related to the absorption of energy by the spin system.

dX''/dB , it refers to the first derivative of the imaginary part of the susceptibility X'' with respect to the magnetic field strength B . This derivative is often used in EPR spectroscopy to analyze the lineshape of the absorption signal, as it can provide information about the relaxation processes and the environment of the unpaired electrons. Second derivative would explicitly require a second differentiation step: d^2X''/dB^2 . Thus, the notation dX''/dB does not represent the second derivative.

11.6. "suggests that red-light irradiation produced electrons." Irradiation with light triggers electron transition between the valence and the conduction band, and formation of a hole-electron pair. A photogenerated electron/hole may be imagined as an open-shell species localized on carbon/nitrogen atom, respectively. As such, irradiation with light does not "produce electrons", but create an electronically excited species, which is a biradical (singlet or triplet in nature).

Response: We thank the reviewer's valuable advice. We have revised the relevant descriptions in the manuscript.

11.7. "electron (e⁻) sacrifier (AgNO₃ or KI) or a hole (h⁺) sacrifier". Scavenger in both instances.

Response: We thank the reviewer's valuable advice. We have revised this mistake in the manuscript.

11.8. Typos in authors' names and papers' titles in references. In particular, in reference 1, 43, 44, 49, 53, 54, 67, 68, 71, 72, 76, 78, 81, 94, etc.

Response: We thank the reviewer's valuable advice. We have checked and updated reference in the manuscript (see below):

Ref. 1: Dorel, R., Grugel, C. P., Haydl, A. M. The Buchwald–Hartwig amination after 25 years. *Angew. Chem. Int. Ed.* **58**, 17118 (2019).

Ref. 43: Tay, N. E. S., Ryu, K. A., Weber, J. L., Olow, A. K., Cabanero, D. C., Reichman, D. R., Oslund, R. C., Fadeyi, O. O., Rovis, T. Targeted activation in localized protein environments via deep red photoredox catalysis. *Nat. Chem.* **15**, 101 (2023).

Ref. 44: Ravetz, B. D., Tay, N. E. S., Joe, C. L., Sezen-Edmonds, M., Schmidt, M. A., Tan, Y., Janey, J. M., Eastgate, M. D., Rovis, T. Development of a platform for near-infrared photoredox catalysis. *ACS Cent. Sci.* **6**, 2053 (2020).

Ref. 49: Mei, L., Moutet, J., Stull, S. M., Gianetti, T. L. Synthesis of CF₃-containing spirocyclic indolines via a red-light-mediated trifluoromethylation/dearomatization cascade. *J. Org. Chem.* **86**, 10640 (2021).

Ref. 53: Ong, W.-J., Tan, L.-L., Ng, Y., H., Yong, S.-T., Chai, S.-P. Graphitic carbon nitride (g-C₃N₄)-based photocatalysts for artificial photosynthesis and environmental remediation: Are we a step closer to achieving sustainability? *Chem. Rev.* **116**, 7159 (2016).

Ref. 54: Savateev, A., Ghosh, I., König, B., Antonietti, M. Photoredox catalytic organic transformations using heterogeneous carbon nitrides. *Angew. Chem. Int. Ed.* **57**, 15936 (2018).

Ref. 67: Cavedon, C., Madani, A., Seeberger, P. H., Pieber, B. Semiheterogeneous dual nickel/photocatalytic (thio)etherification using carbon nitrides. *Org. Lett.* **21**, 5331 (2019).

Ref. 68: Wang, K., Tong, M., Yang, Y., Zhang, B., Liu, H., Li, H., Zhang, F. Visible light-catalytic hydroxylation of aryl halides with water to phenols by carbon nitride and nickel complex cooperative

catalysis. *Green Chem.* **22**, 7417 (2020).

Ref. 71: Khamrai, J., Indrajit Ghosh, I., Savateev, A., Antonietti, M., S., Konig, B. Photo-Ni-dual-catalytic C(sp²)-C(sp³) cross-coupling reactions with mesoporous graphitic carbon nitride as a heterogeneous organic semiconductor photocatalyst. *ACS Catal.* **10**, 3526 (2020).

Ref. 72: Filippini, G., Longobardo, F., Forster, L., Criado, A., Carmine, G. D., Nasi, L., Agostino, C., Melchionna, M., Fornasiero, P., Prato, M. Light-driven, heterogeneous organocatalysts for C-C bond formation toward valuable perfluoroalkylated intermediates. *Sci. Adv.* **6**, eabc9923 (2020).

Ref. 76: Song, H.-Y., Jiang, J., Wu, C., Hou, J.-C., Lu, Y.-H., Wang, K.-L., Yang, T.-B., He, W.-M. Semi-heterogeneous g-C₃N₄/NaI dual catalytic C-C bond formation under visible light. *Green Chem.* **25**, 3292 (2023).

Ref. 78: Zhang, G., Li, G., Lan, Z.-A., Lin, L., Savateev, A., Heil, T., Zafeiratos, S., Wang, X., Antonietti, M. Optimizing optical absorption, exciton dissociation, and charge transfer of a polymeric carbon nitride with ultrahigh solar hydrogen production activity. *Angew. Chem. Int. Ed.* **56**, 13445 (2017).

Ref. 81: Goettmann, F., Fischer, A., Antonietti, M., Thomas, A. Chemical synthesis of mesoporous carbon nitrides using hard templates and their use as a metal-free catalyst for friedel-crafts reaction of benzene. *Angew. Chem. Int. Ed.* **45**, 4467 (2006).

Ref. 98: Johnson Humphrey, E. L. B., Kennedy, A. R., Sproules, S., Nelson, D. J. Evaluating a dispersion of sodium in sodium chloride for the synthesis of low-valent nickel complexes. *Eur. J. Inorg. Chem.* **2022**, e202101006 (2022).

Responses to the Comments by Reviewer: 2

General Comment:

In the manuscript entitled “General Method for Carbon-Heteroatom Cross-Coupling Reactions via Semiheterogeneous Red-Light Metallaphotocatalysis” the authors applied heterogeneous polymeric carbon nitride (CN-OA-m) under the red light illumination for driving the nickel-catalyzed cross-coupling reactions. This catalytic system enabled the formation of four different types of carbon-heteroatom bonds (C-N, C-O, C-S, and C-Se) with a wide range of substrates (more than 200 examples). This study explores an interesting area of heterogeneous photocatalytic for cross-coupling reactions, although there are several previous literatures reported the cross-coupling reactions by using

carbon nitride. However, the manuscript lacks a comprehensive explanation regarding why the material exhibits optimal activity under red light than blue light, yet shows no activity when exposed solely to red light, requiring the combination of red light and heating for activation. Additionally, the observed high yields (>50%) in several other carbon nitride materials such as C_3N_4 and mpg-CN that do not respond to red light further complicate the assertion that the reaction is truly driven by red light. As a result, I find it difficult to fully accept that red light is the primary driving force behind the reaction. Consequently, the proposed reaction mechanism under red light requires further experimental validation and supporting data. In its current form, the manuscript should not be considered for publication in Nature Communications.

Response: We thank the reviewer's valuable advice. The manuscript has been carefully revised based on your comments and suggestions.

Specific Comment

1. In Table 1, materials such as C_3N_4 , mpg-CN, p- C_3N_4 , and g- C_3N_4 are generally considered to lack infrared responsiveness. However, the high activity observed under red light in this study, with yields consistently exceeding 50%, is puzzling. What accounts for this phenomenon? Based on these results, I am inclined to question the assertion that the reaction is driven by red light.

Response: We really appreciate the reviewer's valuable advice. In our study of these photocatalysts (p- C_3N_4 , g- C_3N_4 , mpg- C_3N_4 , C_3N_4 , PR- C_3N_4 , MC- C_3N_4 , CN-OA-m), it was found that PR- C_3N_4 , MC- C_3N_4 , and CN-OA-m exhibited very pronounced UV absorption (Figure S16), while p- C_3N_4 , g- C_3N_4 , mpg- C_3N_4 , and C_3N_4 show relatively weaker red light absorption compared to other catalysts, but still possess a certain degree of light absorption (Figure S17). This slight absorption enables the catalysts to be excited, thereby facilitating the progression of the reactions (See below).

Additionally, we have investigated several of the catalysts used in this paper (C_3N_4 , p- C_3N_4 , g- C_3N_4 , mpg- C_3N_4 , RP- C_3N_4 , MC- C_3N_4 , CN-OA-m) for their property and structural feature, including, XPS spectra, photoluminescence (PL), photocurrent responses, electrochemical impedance spectra, Time-resolved photoluminescence (TRPL) spectra, band structures (see below):

(1) Among these different semiheterogeneous photocatalysts, p- C_3N_4 , g- C_3N_4 , mpg- C_3N_4 demonstrate notably strong photoluminescence (PL) intensity, indicating severe recombination of

photogenerated charge carriers. In contrast, the PL intensity of C₃N₄, RP-C₃N₄, MC-C₃N₄, CN-OA-m exhibits a sequential decrease, implying suppressed recombination of charge carriers. This trend is most pronounced in CN-OA-m, which shows the weakest PL intensity, suggesting an efficient electron transfer from the emissive state to Ni(II) species. This process likely facilitates the formation of Ni(I) species (See below, and please also refer to SI Figure S22) (ref: *Angew. Chem. Int. Ed.* **2017**, *56*, 13445; *Nat Commun.* **2023**, *14*, 1501).

(2) Time-resolved photoluminescence monitored at the corresponding emission peaks give the mean radiative lifetimes (τ) of the recombining charge carriers. The τ of p-C₃N₄, g-C₃N₄, mpg-C₃N₄, C₃N₄, RP-C₃N₄, MC-C₃N₄, CN-OA-m were 1.52, 2.04, 2.48, 2.78, 3.62, 3.73, and 3.88 ns, respectively (for lifetime components see below, please also refer to SI Figure S23, Table S14). The elongation of fluorescence lifetime indeed signifies a decrease in the recombination rate of photogenerated electron-hole pairs, thereby affording photogenerated carriers an extended duration to engage in photocatalytic reactions. This is beneficial for photocatalytic processes, as highlighted by the notably prolonged residual fluorescence lifetime observed in the CN-OA-m. This extended lifetime implies that photogenerated carriers are effectively sustained throughout the reaction, consequently augmenting the efficiency of the photocatalytic reaction (ref: *Nat Commun.* **2023**, *14*, 1501).

(3) The transient photocurrent response of different C₃N₄ electrodes were measured to gain in-depth insight into charge transport (See below, please also refer to SI Figure S24). Compared with other C₃N₄, the CN-OA-m photoelectrode has a strong photocurrent response, indicating the improved separation and transfer of electrons and holes. Similar results were obtained in electrochemical impedance spectroscopy (EIS) measurements for CN-OA-m (See below, please also refer to the SI Figure S25). The arc radius of the CN-OA-m is significantly smaller than that of other C₃N₄, signifying a lower resistance to interfacial charge transfer for the enhanced diffusion mobility of electrons from the CN-OA-m, indicative an efficient electron transfer from the emissive state to Ni(II), resulting the Ni(I) species (*Angew. Chem. Int. Ed.* **2017**, *56*, 13445; *Nat Commun.* **2023**, *14*, 1501).

(4) In addition, the electrochemical potential diagrams of several catalysts indicate that CN-OA-m has the highest reduction potential (refer to SI Figure S26) and exhibits high photogenerated electron-hole pair separation efficiency, which facilitates the photocatalytic coupling reaction. This is why it serves as the best photocatalyst for the reaction examined in this study. the properties of these catalysts align with the yield trend of this photocatalytic reaction, further indicating that the reaction is

a coupling method driven by red light. The corresponding studies and discussion have been supplemented in the manuscript.

Figure S16. Absorption spectrum of different C_3N_4 .

Figure S17. Absorption spectrum of p- C_3N_4 , g- C_3N_4 , mpg- C_3N_4 , and C_3N_4 .

Figure S22. The photoluminescence (PL) spectra of different C_3N_4

Figure S23. Time-resolved photoluminescence (TRPL) spectra of different C_3N_4 .

Table S14. Parameters of fluorescence decay curves for C_3N_4 -OA-m, mpg- C_3N_4 , MC- C_3N_4 , RP- C_3N_4 , C_3N_4 , p- C_3N_4 , g- C_3N_4 samples

Sample	τ_1 (ns) (A ₁ %)	τ_2 (ns) (A ₂ %)	τ_{av} (ns)	Ex (nm)	Em (nm)
CN-OA-m	2.81(97.86)	13.87 (2.14)	3.88	340	533
mpg- C_3N_4	1.86 (98.26)	9.43 (1.74)	2.48	340	457
MC- C_3N_4	2.42(93.81)	9.03 (6.19)	3.73	340	440
RP- C_3N_4	2.33(93.87)	8.82 (6.13)	3.62	340	455
C_3N_4	2.15(98.40)	10.70(1.60)	2.79	340	471

p-C ₃ N ₄	1.19(97.93)	5.11 (2.07)	1.52	340	474
g-C ₃ N ₄	1.43(95.63)	5.49 (4.37)	2.04	340	469

$$\tau_{av} = \frac{A_1\tau_1^2 + A_2\tau_2^2}{A_1\tau_1 + A_2\tau_2}$$

Figure S24. Photocurrent response of different C₃N₄.

Figure S25. The electrochemical impedance spectra (EIS) of different C₃N₄ electrodes.² In Table 1, CN-OA-m, when applied to photocatalytic HER (refer to: *Angew. Chem. Int. Ed.* 56, 13445 (2017)), shows significantly higher activity under blue light compared to green and red light. Why, then, does the activity under red light appear to be lower in this case? What is the underlying cause?

Response: We thank the reviewer's valuable advice. We believe that the significant differences in the photocatalytic hydrogen evolution reaction (HER) activity of CN-OA-m at different light wavelengths can be attributed to several key factors primarily related to the absorption characteristics of the photocatalyst and photon energy:

i) CN-OA-m may exhibit stronger absorption in the blue light region, indicating its ability to capture more photons more effectively within this range. In contrast, red light, with its longer wavelength and lower photon energy, may not be effectively absorbed by the catalyst, resulting in lower photocatalytic activity under red light;

ii) The energy of red light photons is lower than that of blue light photons. If the photocatalytic process requires a certain activation energy, the lower energy of red light photons may be insufficient to effectively excite the electrons in the catalyst, thereby preventing the desired hydrogen evolution reaction (HER) from occurring.

3. When using CN-OA-m under blue light source, but without heating, what is the observed yield?

Response: We thank the reviewer's valuable advice. Following the suggestion, we conducted several coupling reactions between aryl halides and nucleophiles (see below), but unfortunately, no reactions occurred without heating.

^a Reaction conditions: 5-Bromo-*m*-xylene (0.2 mmol), NuH (0.6 mmol), CN-OA-m (10 mg mL⁻¹), *d*-tbbpy (10.0 mol%), NiBr₂·glyme (10.0 mol %), DMAc (1 mL), *m*DBU (1.5 equiv.), blue light (455 nm), r.t., Ar, 24 h.

^a Reaction conditions: Methyl 4-bromobenzoate (0.2 mmol), NuH (0.4 mmol), CN-OA-m (10 mg mL⁻¹), *d*-tbbpy (10.0 mol%), NiBr₂·glyme (10.0 mol %), DMAc (1 mL), *m*DBU (1.5 equiv.), blue light (455 nm), r.t., Ar, 24 h.

4. Previous literature reports that carbon nitriles such as mpg-CN and CN-V, when coupled with Ni catalysts under blue light, can successfully drive C-N coupling reactions (refer to Science 365, 360 (2019) and Cell Rep. Phy. Sci. 2, 100491 (2021)). Although red light is employed in your study, heating is also required. How does this approach demonstrate an advanced or novel aspect of your work?

Response: We thank the reviewer's valuable advice. Transition metal-catalyzed palladium, copper, and nickel Buchwald-Hartwig amination and Ullmann amination reactions provide an efficient method for synthesizing fine chemicals in organic synthesis. However, these reactions often require optimization of reaction conditions on a case-by-case basis. Combining transition metal catalysts with established photocatalysts, such as metal complexes, organic dyes, and organic semiconductors, has emerged as a valuable approach to overcoming these challenges. However, the light-promoted reactions reported to date require blue or high-energy near-UV light, which leads to problems with scalability, chemoselectivity, and catalyst deactivation resulting from competitive absorption of light by the substrates and intermediates. Compared to blue light, the primary advantage of red light in

photocatalysis is its deeper penetration capability (allowing for amplification of the reaction), which allows for amplified reactions. And is able to promote the reaction more efficiently in some cases, and is particularly suitable for aryl halides that can photo-absorb short wavelengths, preventing dehalogenation and deactivation, such as highly π -conjugated molecules (naphthalene, pyrene, etc.) as well as substrates that have photosensitivities (Benzophenone, etc.), thereby minimising competition with the absorption of substrates. The literature primarily focuses on highly reactive aryl iodides (*Cell Rep. Phys. Sci.* **2021**, 2, 100491) and a limited number of electron-deficient aryl bromides (see *Science* **2019**, 365, 360). They have confirmed the feasibility of heterogeneous catalysis in organic synthesis; However, the current range of substrate applications remains limited.

i) In our study, a wide range of substrates is used, including electron-rich, electrically neutral, highly π -conjugated molecules (such as naphthalene, pyrene, etc.), photosensitive substrates (such as benzophenone, etc.), as well as biologically active aryl halides. Moreover, it shows relatively good compatibility with eleven different types of nucleophiles, enabling four different C-heteroatom bonding reactions (C-N, C-O, C-S, C-Se), with more than 200 examples achieving yields up to 94%. This is difficult to achieve in other current methods. The reaction can be carried out on a gram scale in an ordinary round-bottomed flask, which is one of the advantages of red light.

ii) In our previous studies, we found that Ni(I) oxidative addition requires a specific temperature (ref: *J. Org. Chem.* **2022**, 87, 10285; *Chem. Eur. J.*, **2023**, 29, e202300458). Additionally, through density functional theory (DFT), we found that the activation energy required for oxidative addition of electron-rich aryl bromides is 41.5 kJ/mol, indicating it is quite endothermic (see below, B). This further suggests that temperature is crucial in oxidative addition involving Ni(I) species. Our work presents a novel approach for a wide range of C-heteroatom bond coupling reactions, driven by the synergism of light and heat. This may offer new insights into the synergistic effects of light and heat, providing new perspectives for further development of photocatalyst design and optimization of the reaction.

B. DFT calculation

5. If only red light is used for excitation without heating, resulting in no activity, can we conclude that the CN-OA-m does not actually respond to red light? Does this imply that the system should not be classified as a photocatalytic system? Is the red light here primarily acting in a thermal capacity, but possibly differing from direct heating? What is the specific mechanism behind this behavior? This uncertainty in the reaction mechanism requires further experimental validation to confirm whether the reaction is truly driven by red light. This remains a significant issue in the manuscript.

Response: We thank the reviewer's valuable advice. According to our mechanistic studies reveal that without heating, the reaction is inactive under red light excitation (<40 °C). Instead, red light excitation of CN-OA-m reduces Ni(II) to Ni(I) species under room temperature. This was further confirmed by EPR experiments, which detected a signal showing the generation of Ni(I) species (see below, C). We synthesized a stable Ni(I) species, which shows almost no reactivity when used as a catalyst at temperatures below 40°C; only by increasing the temperature can the reaction proceed (see below, D). Meanwhile, in our earlier studies, we discovered the Ni(I) species requires a certain temperature for oxidative addition with aryl halides, which explains why the reaction does not occur under red light excitation at room temperature. (ref: *J. Org. Chem.* **2022**, *87*, 10285; *Chem. Eur. J.*, **2023**, *29*, e202300458). Additionally, through density functional theory (DFT), we found that the activation energy required for oxidative addition of electron-rich aryl bromides is 41.5 kJ/mol (see below, B), indicating it is quite endothermic (see below, B). This further suggests that temperature is crucial in oxidative addition involving Ni(I) species. The catalytic cycle is then completed through Ni(I)-Ni(III), which also demonstrates that our reaction is a process driven by the synergy of light and

heat.

C. Detection of Ni species generation under room temperature conditions.

D. Ni(I) species as a catalyst initiates the C-N coupling reaction.

B. DFT calculation

6. In Table S6, the trends observed for different light sources are quite unusual. The catalytic effect of green LED (520 nm–530 nm) is higher than that of the 580-595 nm light source, yet when using a 620 nm–630 nm source, the yield exceeds that of the green LED. What explains this unexpected trend?

Response: We thank the reviewer's valuable advice. Based on the reviewers' suggestions, we examined the 580-595 nm light source and found that the existing 580-595 nm LED beads have a luminous angle of 150 degrees and an optical power of 80 mW/cm², which may be the reason for the anomalies in our data. In our earlier studies (ref. *ACS Catal.* **2024**, *14*, 4968), we discovered that light intensity significantly affects the reaction yields. Therefore, we repurchased 580-595 nm LED beads with a luminous angle of 45 degrees (currently, the emission angle for red light is 45 degrees) and an optical power of 360 mW/cm². We conducted experiments again using this 580-595 nm light source (see below), and the new data has now been added to the supplementary information.

7. Based on the DRS and Figure S25, it is apparent that Ni particles are deposited on CN-OA-m. Furthermore, the relative ratio of the two peaks at 166 and 159 in the NMR spectrum shows a significant change. Shouldn't this indicate substantial structural alterations? Therefore, the claim of minimal structural change seems inconsistent. What caused this structural change observed in the NMR spectra?

Response: We thank the reviewer's valuable advice. In our study of the CN-OA-m cycle recycling experiments, we found that the catalyst exhibited high activity during the first five cycles. Characterization of the CN-OA-m features during these initial cycles indicated that its structural changes were relatively stable. The relative ratio of the two peaks at 166 and 159 in the NMR spectrum showed variations, likely due to the number of scans during testing. We have retested the samples (increased the number of scans to 8000) (see below, SI Figure S37) and updated the NMR data in the supplementary information. Furthermore, in the sixth cycle, we observed a decline in the activity of CN-OA-m, with 47% of aromatic amine compounds obtained after the seventh recovery (see below). Through HAADF-STEM (Figure S33), we detected the formation of deposited nickel aggregates on the surface of the recovered CN-OA-m, probably embedded Ni into the six-fold cavity in carbon nitride through the coordination between Ni and N-conjugated aromatic ligands of carbon nitride, which cannot form Ni (I) active species. Due to potentially misleading statements in our previous presentation, which may have caused misunderstandings for readers, we have updated the manuscript and included relevant discussions.

Figure S35. Solid-state ^{13}C NMR spectra new (left) and recovered (right).

Research on the recycling and reuse of

CN-OA-m.

Figure S11. Recycling experiment performed under standard condition for 24 hours.

8. The captions for Figures S21-S27 in the Supplementary Materials are all incorrect.

Response: We thank the reviewer's valuable advice. We have revised the relevant mistake in the SI.

9. What are the conduction band and valence band positions of CN-OA-m? Does it possess sufficient reduction potential to reduce $\text{Ni}^{\text{II}}/\text{Ni}$ (-1.43 V vs SCE)?

Response: We thank the reviewer's valuable advice. The conduction band position of CN-OA-m is

-1.65 V, and the valence band is at 0.88 V (ref: *Nat. Catal.* **2020**, 3, 611). It possesses a sufficient reduction potential to reduce Ni(II) to Ni(I) species.

10. In Figure 1C, the EPR results may be inaccurate. According to previous studies using EPR to characterize carbon nitrides, a distinct peak for unpaired electrons around $g=2.0$ should be observed under dark conditions even with standalone carbon nitride (see *Angew. Chem. Int. Ed.*, 2019, 58(11): 3433-3437). However, this peak is absent here, which is inconsistent. This cannot serve as evidence for infrared responsiveness.

Response: We thank the reviewer's valuable advice. The structure in the literature (ref: *Angew. Chem. Int. Ed.* **2019**, 58, 3433) may differ somewhat from our structure. Through multiple electron paramagnetic resonance (EPR) experiments, we found that under dark conditions, there were only a very weak signal peaks near $g=2.0$ for CN-OA-m (See below). However, under stimulated excitation with red light, a significant signal appeared, indicating that CN-OA-m generates a greater number of hole-electron pairs (create an electronically excited species, which is a radical). Additionally, the ultraviolet-visible absorption spectrum of CN-OA-m does respond to red light in the range of 500-700 nm, indicating that CN-OA-m can be utilized as a red light catalyst. Meanwhile, we have investigated several catalysts (C_3N_4 , p- C_3N_4 , g- C_3N_4 , mpg- C_3N_4 , RP- C_3N_4 , MC- C_3N_4 , CN-OA-m) for their property and structural feature, including photoluminescence (PL), fluorescence lifetime, photocurrent responses, electrochemical impedance spectroscopy and band structures. These studies indicate that the porous CN-OA-m exhibits a high efficiency in the separation of photogenerated electron-hole pairs, making it a suitable catalyst for facilitating the transfer of electrons from the excited state to Ni(II), resulting in the formation of Ni(I) species.

11. In Figure 1E, the EPR results are also problematic. Why is there no signal near $g=2.0$ for CN-OA-m? Additionally, EPR peaks for Ni(I) typically show splitting, yet this is not observed here, which is unreasonable (see *Angew. Chem. Int. Ed.*, 2023, 62(43): e202310753).

Response: We thank the reviewer's valuable advice. In EPR experiments, different detection conditions must be selected based on the specific testing objectives. Under room temperature conditions, EPR experiments primarily focus on detecting free radical signals. Since Ni(I) species are nearly undetectable at room temperature, testing must be conducted at low temperatures to obtain hyperfine coupling data. Consequently, at 100K, the hole signal of CN-OA-m cannot be observed near $g=2.0$. Furthermore, the splitting of Ni(I) species is often significantly influenced by the testing environment, such as temperature and solvent. Our earlier studies indicated that Ni(I) species exhibit splitting when toluene is used as a solvent (see below, the left figure, ref: *Chem. Eur. J.*, **2023**, 29, e202300458;), while our recent study found that Ni(I) species do not show splitting in DMSO (ref: *Angew. Chem. Int. Ed.* **2024**, e202314355; see below, the right figure). These findings further illustrate that the splitting of Ni(I) species is significantly affected by environmental factors.

Responses to the Comments by reviewer: 3

General Comment:

Compared to molecular transformation reactions using blue-to-green light, red-to-near-infrared (NIR) light has clear advantages, but such research has been limited. A key challenge has been the development of catalysts capable of effectively interacting with red-to-NIR light while exhibiting appropriate reactivity with substrates. The authors propose carbon nitride-type heterogeneous semiconductors as a novel class of red-light photocatalysts to address this issue. Specifically, the authors focus on the recently reported CN-OA-m catalyst and its application to red-light-driven nickel-catalyzed carbon-heteroatom cross-coupling reactions. One of the primary advantages of this reaction is its broad applicability to a wide range of nucleophiles without case-by-case optimization of the reaction conditions. Additionally, it leverages the benefits of heterogeneous catalysis, demonstrating recyclability. The reaction mechanism is thoroughly analyzed with appropriate references to previous findings. While these findings underscore the potential of CN-OA-m as a novel red-light photocatalyst, the advantages of utilizing red light were not sufficiently addressed. My concerns are outlined below. If these aspects are adequately clarified, this paper will be one of the standards in the chemistry of red-light-driven photocatalysts.

Response: We thank the reviewer's valuable advice. The manuscript has been carefully revised based on your comments and suggestions.

Specific Comment

1. General reaction conditions applied to various substituents and nucleophiles represent a significant advantage for this reaction. This reaction proceeds via two catalytic cycles as the authors described (Scheme 5): one involving CN-OA-m photocatalyst and red light and another involving a nickel catalyst. While a broad range of nucleophiles can be used, their reactivity is governed primarily by the latter catalytic cycle. Since this cycle does not require light, the observed advantage may not be unique to the CN-OA-m photocatalyst. Could some nickel complex intermediates in Scheme 5 absorb shorter-wavelength light and compete with undesired reactions? If this hypothesis is valid, can the authors provide spectroscopic data on the intermediates to support it?

Response: We thank the reviewer's valuable advice. In the catalytic cycle of Scheme 6, the

CN-OA-m catalytic cycle involving red light excitation and nickel catalysis is a complementary process. Red light plays a crucial role in the reaction. Without the excitation of CN-OA-m by red light, the reaction cannot successfully produce Ni(I) species, and there will be no subsequent nickel catalytic cycle. During the nickel catalytic cycle, there may exist dtbbpy-Ni(I)-Br or dtbbpy-Ni(III)-Br₂(Ar) species, which could exhibit light absorption in the short wavelength region (dtbbpy-Ni(II)-Br₂ has light absorption in the ultraviolet range of 360-390 nm). Unfortunately, in our catalytic system, due to the extreme instability of dtbbpy-Ni(I)-Br or dtbbpy-Ni(III)-Br₂(Ar) species, we have not been able to obtain effective data despite multiple ultraviolet experiments. Additionally, we only detected Ni(I) species at a temperature of 100K through EPR experiments, while we were unable to detect them at room temperature. Meanwhile, in the nickel catalytic cycle, the simultaneous presence of Ni(I) and Ni(III) species can lead to the loss of catalyst activity, resulting in the transformation into inert dtbbpy-Ni(II)-X₂ species. In our previous studies (ref: *Chem. Eur. J.*, **2023**, *29*, e202300458), the dtbbpy-Ni(II)-X₂ complex was unable to successfully generate Ni(I) species under purple light excitation. Only by adding an organic base as an electron donor could the reduction from Ni(II) to Ni(I) species be achieved under purple excitation.

2. The authors described that this reaction requires light and heat (85°C). If the proposed reaction mechanism is correct, heat is essential for the nickel-catalyzed process. The nickel-catalyzed process should have been previously reported, and a more detailed discussion of the role of heat is necessary.

Response: We thank the reviewer's valuable advice. According to the reviewer's suggestion, we have studied the effect of different temperatures on reaction yield and found that temperature is indeed critically important for the reaction to proceed. At temperatures below 40 °C, the reaction hardly proceeds. As the temperature increases, the yield increases significantly, but beyond 90 °C, further increases in temperature lead to negligible changes in yield (see below, SI Section 6). Meanwhile, in our earlier studies on Ni chemistry, we discovered that heating exerts a pronounced effect on the oxidative addition of Ni(I) species with aryl halides (ref: *J. Org. Chem.* **2022**, *87*, 10285; *Chem. Eur. J.*, **2023**, *29*, e202300458). Additionally, through density functional theory (DFT), we determined that the activation energy for the oxidative addition of electron-rich aryl bromides is 41.5 kJ/mol, indicating it is quite endothermic (see below, B). This further suggests that temperature is crucial in oxidative addition involving Ni(I) species. We have added the corresponding discussion in the manuscript.

A. Effect of temperature on reaction under red light excitation

B. DFT calculation

3. What temperature were the measurements in Figure 1E conducted? Can the effect of heat be further discussed through temperature-dependent measurements?

Response: We thank the reviewer's valuable advice. The electron paramagnetic resonance (EPR) experiment depicted in Figure 2E was conducted to detect Ni(I) species under conditions of 100K. The experimental procedure involved thoroughly mixing CN-OA-m with Ni(II) species before adding it to a DMF solution. A small amount of the sample was then placed directly into a EPR tube. After irradiating with a red LED light for EPR tube, we concluded that Ni(I) species had been generated.

Subsequently, the EPR tube containing the sample was rapidly frozen in liquid nitrogen, and the EPR spectrum was measured at 100K. Therefore, the process was essentially free from temperature effects.

4. While the recycled catalysts have been characterized using various measurements, can the authors also include XRD analysis?

Response: We thank the reviewer's valuable advice. We have added the XRD analysis of recycled catalysts (see below) in the Supporting Information.

Figure S36. XRD patterns new (left) and recovered (right).

5. The authors proposed that *m*DBU is the optimized base to react with the hole of the catalyst. Is the oxidation potential of *m*DBU consistent with this role? Can the authors discuss the differences in oxidation potentials of bases presented in Table S7?

Response: We thank the reviewer's valuable advice. We tested the redox potentials of various organic bases (See below), the oxidation potentials of DBU, *m*DBU, MTBD, *t*Bu-TMG, TMG, TBD, DIPEA, Et₃N, Cy₂NMe, DBN, quinuclidine, and DABCO are 1.81V, 2.14V, 1.27V, 1.71V, 1.82V, 0.96V, 1.65V, 1.36V, 0.94V, 2.03V, 1.44V, and 1.01V, respectively. And found that the oxidation potential of *m*DBU is 2.14V, which has a better matching oxidation potential. Relevant discussions and data have been added to the manuscript and supporting information.

6. The broad scope of this reaction to a wide range of substrates and nucleophiles is highly valuable. However, most of the substrates can interact with only UV light. One of the key advantages of red-light photocatalysis, as noted by the authors, is that it minimizes competition with the absorption of substrates. Therefore, I strongly recommend the authors demonstrate the applicability of this reaction to substrates that strongly absorb visible light, such as highly π -conjugated molecules.

Response: We thank the reviewer's valuable advice. We have investigated a series of highly π -conjugated molecules and photosensitive aryl halides, including naphthalene, anthracene, and pyrene (see below). We have added the relevant discussions in the manuscript (Scheme 2).

7. The reactions presented in this manuscript settle on one reaction mechanism. Further exploration of CN-OA-m's applicability to other photochemical reactions should be valuable, while many examples are not required. Such results can provide crucial insights into whether conventional photochemical reactions can be adapted for red-light-driven reactions using this catalyst.

Response: We thank the reviewer's valuable advice. We have conducted some preliminary explorations of CN-OA-m as a red light catalyst for Minisci-type photochemical reactions (as detailed below). Fortunately, we currently have reactions with preliminary results, and our laboratory is further investigating these specific studies.

We believe the revision has addressed most, if not all, of the reviewers' concerns, and we look forward to hearing from you.

Thank you for your consideration and best regards,

Yours sincerely,

Dong Xue

Responses to the Comments by Reviewer: 1

General Comment:

In this revised version of the manuscript, authors addressed satisfactory ~80% of my original comments. Comments from the original report that require further improvements are:

Specific Comment

1. Please report specific surface areas of carbon nitrides that were obtained from N₂ physisorption measurements.

Response: We thank the reviewer's valuable advice. We have supplemented the surface areas of different carbon nitrides in the SI (See below, Table S14).

Table S14. BET specific surface area, pore diameter and total pore volume.

Sample	Specific surface area (m ² g ⁻¹)	Pore diameter (nm)	Total pore volume (ccg ⁻¹)
CN-OA-m	43.80	13.80	0.003037
C ₃ N ₄	63.70	11.47	0.002461
p-C ₃ N ₄	11.13	24.05	0.001051
g-C ₃ N ₄	7.70	28.12	0.000792
mpg-C ₃ N ₄	174.96	9.83	0.005687
RP-C ₃ N ₄	2.45	17.17	0.000985
MC-C ₃ N ₄	2.14	13.44	0.000152

2. Section "Results and discussion", first sentence "conduction band"  "conduction band potential" and "valence band"  "valence band potential". Also, please indicate the reference electrode.

Response: We thank the reviewer's valuable advice. We have corrected the relevant descriptions in the manuscript.

3. Based on group's earlier publication and considering redox potentials, authors assumed that *m*DBU is oxidized. Attempts to detect and/or isolate any products of *m*DBU oxidation were not made.

Response: We appreciate the reviewer's valuable advice. Based on their recommendations, we conducted the following study: in standard reactions, TEMPO was added as a capture reagent, and we detected the adduct formed between the cationic radical of *m*DBU and TEMPO using high-

resolution mass spectrometry (HR-MS). This result preliminarily confirms that mDBU was oxidized during the reaction. The main process involves the photoinitiation of CN-OA-m under red light, where electrons in the valence band are excited to the conduction band, forming electron-hole pairs. The electrons in the conduction band reduce Ni(II) complexes to Ni(I) species through single-electron transfer, while the holes in the valence band oxidize the organic base (mDBU), generating cationic radicals and simultaneously restoring the photocatalyst to CN-OA-m.

In addition, a few questions arose after reading the revised manuscript:

4. Authors measured oxidation potential of bases. In particular, they found that oxidation potential of

*m*DBU (presumably peak) is +2.14 V vs. Ag/AgCl. Given the potential of the valence band of CN-OA-m of + 0.8 V indicated in the Results and Discussion section, electron transfer from this reactant to CN-OA-m electronically excited state must be strongly endergonic, and as such does not proceed at significant rate. More correct approach, which is actually adopted in literature, is to report ($E_{p/2}$) potential at half-peak. See for example, *Synlett* **2016**; 27(05): 714-723. This will give a more reasonable value.

Response: We thank the reviewer's valuable advice. We have updated the relevant descriptions in the manuscript based on the $E_{p/2}$ potential correction and added in SI (See below).

5. Discussion of the mechanism, “the as-formed holes (h^+) from the emissive state to reduce Ni(II) (E_i [NiII/NiI] = -1.43 V vs SCE) to generate a Ni(I) species via SET”. Probably authors mean “the as-formed electrons (e^-)”.

Response: We thank the reviewer’s valuable advice. We have corrected the relevant descriptions in the manuscript.

6. Authors corrected some typos, but introduced other new typos.

6.1. Figure 2C. X-axis label “maNgnetic field”. Y-axis label “eRP siNGal”. Also, is there any good reason why authors use different axes labels in Figure 2C and 2E, “B/G” vs. “maNgnetic field” and “dX'/dB” vs. “eRP siNGal”?

Response: We thank the reviewer’s valuable advice. To avoid any misunderstandings for the readers, we have standardized the axes labels in Figure 2.

6.2. Scheme 1, CN-OA-m structure. It appears that one of the bridging N-atom is methylated. Is it a typo?

Response: We thank the reviewer’s valuable advice. We have revised this mistake in the Scheme 1.

6.3. Section title “C–heteroatom couplings”.

Response: We thank the reviewer’s valuable advice. We have revised this mistake in the manuscript.

Overall, the scientific content is of high quality. However, the quality of writing requires substantial improvement. I will leave this aspect to the authors and editor's discretion. If it is acceptable to publish a paper with obvious typos in a flagship journal of the Nature family journals.

Reviewer #2 (Remarks to the Author):

REVIEW 2# — Joint Evaluation with Reviewer 3# Comments

Thank you for inviting me to re-evaluate the revised manuscript entitled “General Method for Carbon–Heteroatom Cross–Coupling Reactions via Semiheterogeneous Red-Light Metallaphotocatalysis” by Song et al. In this resubmission, the authors have made several substantial improvements. Notably, they addressed critical points regarding the advantages of CN-OA-m, broadened the substrate scope, investigated the role of reaction temperature, and incorporated additional characterizations to support their structure–activity relationship analysis. These revisions, along with the correction of previous inaccuracies, have resulted in a clearly enhanced manuscript. This present article may become suitable for publication in Nature Communications. While, several concerns still require clarification:

Specific Comment

7. In Figure S20, the N₂ adsorption/desorption isotherms for MC-C₃N₄ and PR-C₃N₄ are not closed, which is abnormal and suggests potential issues with the measurements. Re-measurement is strongly

recommended.

Response: We thank the reviewer's valuable advice. We have remeasured the N_2 adsorption/desorption isotherms of MC- C_3N_4 and RP- C_3N_4 by increasing the mass (660 mg) and updated the new data in the SI (See below).

Table S14. BET specific surface area, pore diameter and total pore volume.

Sample	Specific surface area ($m^2 g^{-1}$)	Pore diameter (nm)	Total pore volume ($cc g^{-1}$)
CN-OA-m	43.80	13.80	0.003037
C_3N_4	63.70	11.46909	0.002461
p- C_3N_4	11.1311	24.05477	0.001051
g- C_3N_4	7.6994	28.11851	0.000792
mpg- C_3N_4	174.9604	9.83135	0.005687
RP- C_3N_4	2.45	17.17	0.000985
MC- C_3N_4	2.14	13.44	0.000152

8. In Figure S20, the light absorption of p- C_3N_4 , g- C_3N_4 , and C_3N_4 beyond 550 nm is extremely weak. It is difficult to believe that such minimal absorption would lead to any significant photoactivity under red-light irradiation. Moreover, the DRS spectra should include labeled y-axes. It is advised that the authors provide more convincing evidence demonstrating that these materials indeed exhibit red-light-responsive photocatalytic behavior under specific thermal conditions.

Response: We thank the reviewer's valuable advice. Due to the weak light absorption of p- C_3N_4 , g- C_3N_4 , and C_3N_4 above 550 nm, we speculate that a portion of the metallic nickel may be loaded onto carbon nitride during the reaction process, thereby enhancing the light absorption intensity. We conducted

UV absorption experiments on the carbon nitride after the reaction and found that its absorption indeed improved, which might explain the effectiveness of these catalysts (see below up). Additionally, we have labeled the y-axis in the DRS spectra in the SI (See below down).

9. Considering the thermal radiation associated with red-light illumination, it is recommended that the surface temperatures of the catalyst be measured separately under (i) red-light irradiation alone and (ii) combined red-light and thermal heating conditions.

Response: We thank the reviewer's valuable advice. Under the current room temperature conditions, we irradiated the catalyst CN-OA-m with red light for 10 minutes and measured its surface temperature using an infrared thermal imager (*Rev. Sci. Instrum.* 2024, 95, 035116), which was about 35.5 °C (See left). Under heating conditions (85 °C), when irradiated with red light, the surface temperature of the catalyst CN-OA-m was approximately 81.1 °C (see right).

10. In Figure S36, a new XRD peak appears at approximately 40° in the recycled catalyst. What is the

origin of this peak? Further explanation or identification is required.

Response: We thank the reviewer's valuable advice. Due to instrument noise issues, we have updated the new XRD data after multiple measurements.

Reviewer #4 (Remarks to the Author):

For the DFT calculations, the methods and basis sets of geometry optimization and single point energy calculation are acceptable. The solvation effect is also considered to ensure the credibility of the results. However, I disagree with the statement "the activation energy requisite for the oxidative addition of electron-rich aryl bromides amounts to 41.5 kJ/mol, indicating it is quite endothermic. This further suggests that temperature is crucial in oxidative addition involving Ni(I) species.". It is generally believed that reactions with free energy barriers lower than 20 kcal/mol can spontaneously occur at room temperature, For example K. N. Houk* *Journal of the American Chemical Society* 2022 144 (4), 1971-1985.

Response: We sincerely thank the reviewer for their insightful comments. As noted in the cited literature (*J. Am. Chem. Soc.*, **2022**, *144*, 1971), reactions with a free energy barrier below 20 kcal/mol can often proceed spontaneously at room temperature. However, based on our control experiments reveal that temperature significantly influences the reaction outcome, as lower temperatures lead to a substantial decrease in product yield (See below, see SI section 6). These findings further underscore the importance of temperature control in this transformation. To better understand this discrepancy, we are conducting further investigations, including more rigorous DFT calculations, to elucidate the role of temperature in the reaction mechanism. Considering the complexity of photocatalytic systems, these studies may require considerable time to complete. To ensure the rigor of the manuscript and to avoid any potential

misunderstandings, we removed the preliminary speculations regarding DFT calculations from this manuscript. Regarding the more in-depth study of density functional theory (DFT) calculations and mechanisms studies were conducted as independent research work. we particularly appreciate the reviewers for their constructive suggestions, which have significantly contributed to deepening our understanding of the reaction process.

Effect of temperature on reaction under red light excitation

We believe the revision has addressed most, if not all, of the reviewers' concerns, and we look forward to hearing from you.

Thank you for your consideration and best regards,

Yours sincerely,

Dong Xue

Reviewer #1 (Remarks to the Author):

General Comment:

Authors addressed all the remaining issues. The manuscript may, in principle, be recommended for publication in Nature Communications.

Response: Thank you very much for reviewing our manuscript.

Reviewer #2 (Remarks to the Author):

The issues mentioned have already been addressed.

Response: Thank you very much for reviewing our manuscript.